# Plekhg5-regulated autophagy of synaptic vesicles reveals a pathogenic mechanism in motoneuron disease

Patrick Lüningschrör[1,2], Beyenech Binotti[3], Benjamin Dombert[1], Peter Heimann[2], Angel Perez-Lara [3], Carsten Slotta[2], Nadine Thau-Habermann[4], Cora R. von Collenberg[1], Franziska Karl[5], Markus Damme[6], Arie Horowitz [7], Isabelle Maystadt[8], Annette Füchtbauer[9], Ernst-Martin Füchtbauer[9], Sibylle Jablonka[1], Robert Blum[1], Nurcan Üçeyler[5], Susanne Petri[4,10], Barbara Kaltschmidt[2,11], Reinhard Jahn[3], Christian Kaltschmidt[2] & Michael Sendtner [1]

Autophagy-mediated degradation of synaptic components maintains synaptic homeostasis but also constitutes a mechanism of neurodegeneration. It is unclear how autophagy of synaptic vesicles and components of presynaptic active zones is regulated. Here, we show that Pleckstrin homology containing family member 5 (Plekhg5) modulates autophagy of synaptic vesicles in axon terminals of motoneurons via its function as a guanine exchange factor for Rab26, a small GTPase that specifically directs synaptic vesicles to preautophagosomal structures. *Plekhg5* gene inactivation in mice results in a late-onset motoneuron disease, characterized by degeneration of axon terminals. Plekhg5-depleted cultured motoneurons show defective axon growth and impaired autophagy of synaptic vesicles, which can be rescued by constitutively active Rab26. These findings define a mechanism for regulating autophagy in neurons that specifically targets synaptic vesicles. Disruption of this mechanism may contribute to the pathophysiology of several forms of motoneuron disease.

[1] Institute of Clinical Neurobiology, University Hospital Würzburg, 97078 Würzburg, Germany. [2] Department of Cell Biology, University of Bielefeld, 33501 Bielefeld, Germany. [3] Department of Neurobiology, Max Planck Institute for Biophysical Chemistry, 37077 Göttingen, Germany. [4] Department of Neurology, Hannover Medical School, 30625 Hannover, Germany. [5] Department of Neurology, University Hospital Würzburg, 97078 Würzburg, Germany. [6] Institut für Biochemie, Christian-Albrechts-Universität zu Kiel, 24098 Kiel, Germany. [7] Cardeza Vascular Biology Center, Departments of Medicine and Cancer Biology, Sidney Kimmel Medical College, Thomas Jefferson University, Philadelphia, PA 19107, USA. [8] Centre de Génétique Humaine, Institut de Pathologie et de Génétique, 6041 Gosselies, Belgium. [9] Department of Molecular Biology and Genetics, Aarhus University, 8000 Aarhus C, Denmark. [10] Integrated Research and Treatment Center Transplantation (IFB-Tx) Hannover, Hannover Medical School 30625 Hannover, Germany. [11] Molecular Neurobiology, University of Bielefeld, 33615 Bielefeld, Germany. Beyenech Binotti, Benjamin Dombert, Peter Heimann, Christian Kaltschmidt and Michael Sendtner contributed equally to this work. Correspondence and requests for materials should be addressed to C.K. (email: c.kaltschmidt@uni-bielefeld.de) or to M.S. (email: Sendtner_M@ukw.de)

In neurons, autophagosome biogenesis predominantly takes places in axon terminals. Newly synthesized autophagosomes mature during retrograde transport to the soma and fuse with lysosomes[1, 2]. Autophagy-mediated degradation of synaptic components is involved in neuronal network remodeling, presynaptic neurotransmission and synaptic pruning at polyinnervated neuromuscular junctions during development, but could also represent a mechanism of neurodegeneration[3–6].

Selective elimination of axon terminals marks disease onset in familial amyotrophic lateral sclerosis (ALS) and loss of synaptic vesicles precedes axon degeneration in SOD1 G93A mice[7], indicating that autophagy temporally regulates degradation of synaptic vesicles in axon terminals. Neuron-specific disruption of autophagy by depletion of *Atg5* or *Atg7* results in progressive deficits of motor function and neurodegeneration[8, 9]. Modulation of the autophagy pathway modifies disease onset and progression in SOD1 G93A mice, which further points to the contribution of autophagy to the pathology of motoneuron disease[10, 11]. However, the molecular mechanisms that differentially regulate autophagy of synaptic vesicles and components of presynaptic active zones are not known.

Here, we show that Plekhg5 (also known as Syx, Tech or GEF720)[12–14] regulates autophagy of synaptic vesicles in motoneurons via its function as a guanine exchange factor (GEF) for Rab26, a small GTPase selectively delivering synaptic vesicles into preautophagosomes[15]. Plekhg5 is a pleckstrin homology domain containing member of the GEF family. It is predominantly expressed in the nervous system[12, 14]. While several studies provided evidence for a function of Plekhg5 in endothelial cell migration and angiogenesis in zebrafish and mice[13, 16], its function in the nervous system remained elusive. Mutations in the *PLEKHG5* gene have been associated with different forms of motoneuron diseases such as distal spinal muscular atrophy type IV (DSMA-IV)[17], intermediate Charcot-Marie-Tooth disease (CMT)[18, 19] and ALS[20]. The *PLEKHG5*-mutations identified so far in CMT disease patients produce premature stop codons and are predicted to result in a functional null allele[18, 19]. In contrast, mutations described to be causative for a recessive lower motoneuron disease with childhood onset lead to an exchange of one amino acid within the pleckstrin homology domain[17, 20].

To analyze the function of Plekhg5 in the nervous system and its involvement in motoneuron disease, we generated Plekhg5-deficient mice using a gene-trap approach[21]. Plekhg5-deficient mice develop a late-onset motoneuron disease, characterized by degeneration of motoneuron axon terminals. Plekhg5-depleted cultured motoneurons show defective autophagy resulting in accumulation of synaptic vesicles at axon terminals and impaired axon growth. Furthermore, Plekhg5-deficient cells show a reduced activity of Rab26. In a cell-free system Plekhg5 is able to promote GTP exchange of Rab26. Finally, constitutively active Rab26 rescues autophagy and axon growth deficits of cultured motoneurons. Our data indicate that the function of Plekhg5 as a GEF for Rab26 is essential for axonal integrity, and that defective autophagy of synaptic vesicles is a pathogenic mechanism in this specific form of motoneuron disease.

## Results

### Plekhg5 deficiency causes a late-onset motoneuron disease.
To analyze the cellular function of Plekhg5, we generated Plekhg5-deficient mice using an embryonic stem-cell line harboring a gene-trap cassette within the *Plekhg5* allele (Fig. 1), (Supplementary Fig. 1)[21]. Plekhg5-deficient mice developed normally without obvious disease phenotype up to adulthood. However, within 24 months after birth 33 out of 100 homozygous mice died, in contrast to 10 out of 100 heterozygous and 5 out of

100 wild-type animals (Fig. 1a). Affected mice developed hind-limb paralysis at 15 months (Fig. 1b). A reduction in body weight became apparent in 18–24 months old Plekhg5-deficient mice (Fig. 1c). To assess the impact of Plekhg5 deficiency on motor performance in more detail, grip strength (Fig. 1e, f) and rotarod performance (Fig. 1g) were analyzed in animals of different ages. These data indicate that the disease starts late at about 12 months of age and progresses from hindlimbs (Fig. 1f) to forelimbs (Fig. 1e). Correlating with reduced grip strength, we observed a loss of motoneurons starting at 12 months as shown by quantification of Nissl-stained lumbar spinal cord sections (Fig. 1d, h). This decrease was more prominent in 24-month-old animals (Fig. 1d).

We complemented the phenotypic characterization of the motor system with nerve conduction and electromyography studies of the gastrocnemius and plantaris muscles (Fig. 1j–l, o). In both muscles we observed a significantly delayed motor latency time (Fig. 1i). The measurements in the gastrocnemius muscles also showed a marked reduction of motor unit potential (MUP) amplitudes following sciatic nerve-stimulation (Fig. 1j). Our analyses revealed elevated single motor unit potentials (SMUP) (Fig. 1k) and a decrease in motor unit number estimation (MUNE) (Fig. 1l) in the gastrocnemius muscle, suggestive of sprouting of surviving motoneurons to re-innervate muscle fibers that have been denervated upon motoneuron loss. This finding is also supported by the observed fiber grouping as shown by succinate dehydrogenase (SDH) staining (Fig. 1m). These data are similar to the observed sprouting phenotype reported in SOD G93A mice[22]. As previously described for *Plekhg5* mutant mice[19], we also detected a modest decrease in nerve conduction velocities (NCVs) (Fig. 1o).

Next, we investigated β-GEO reporter-activity of the gene-trap cassette in mutant mice. In contrast to spinal cord, β-galactosidase activity was not detected in skeletal muscle (Fig. 1n). Therefore, loss of Plekhg5 apparently does not affect the skeletal muscle directly, but rather indirectly via denervation and motoneuron loss.

We also assessed the impact of Plekhg5 deficiency on sensory and cognitive function. Sensory function was analyzed using the von Frey and Hargreaves tests (Fig. 1p, q). Whereas Plekhg5-deficient mice did not show major changes in mechanical withdrawal thresholds as compared to wild-type littermates (Fig. 1p), they displayed hypersensitivity to heat (Fig. 1p), indicating that sensory perception is not reduced in these mice.

To address whether Plekhg5 deficiency impairs cognitive function, we performed behavioral tests in 12–14-month-old mice (Supplementary Fig. 2), before motor disease symptoms occur. Plekhg5-deficient mice performed similarly in the open field test (Supplementary Fig. 2a, b) indicating that anxiety-related behavior is not altered. They also did not display differences in object recognition (Supplementary Fig. 2c, d) and fear conditioning (Supplementary Fig. 2e, f) suggestive for unaffected learning and memory.

### Plekhg5 modulates synaptic vesicle turnover in motoneurons.
Next, we studied neuromuscular junctions (NMJs) within the gastrocnemius muscle to analyze whether the degenerative process in *Plekhg5*−/− mice involves terminal axons (Fig. 2a, c; Supplementary Fig. 3). Wild-type animals exhibited regular pretzel-like NMJs at an age of 12 months, as indicated by bungarotoxin (BTX) staining for labeling of postsynaptic acetylcholine-receptors and synaptophysin immunoreactivity for the corresponding presynaptic compartment (Fig. 2a; Supplementary Fig. 3). In contrast, the presynaptic nerve

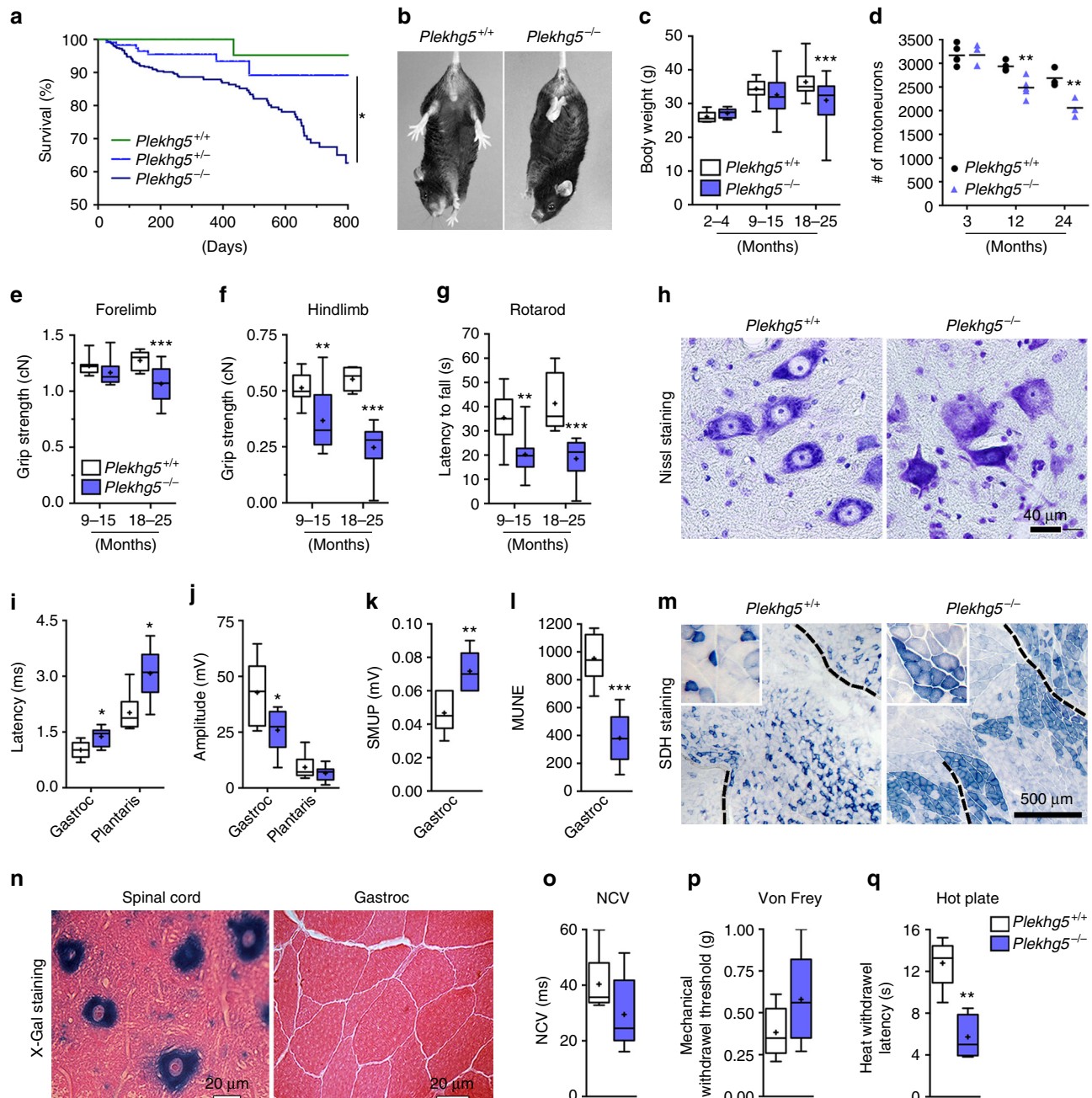

**Fig. 1** Plekhg5-deficient mice develop a motoneuron disease with late onset. **a** Survival analysis of *Plekhg5* mutant mice. (Mean survival after 24 months: *Plekhg5*$^{+/+}$: 95%, $n = 48$; *Plekhg5*$^{+/−}$: 89%, $n = 112$; *Plekhg5*$^{−/−}$: 68%, $n = 351$; $p = 0.0068$; log-rank test) **b** Hind-limb clasping defects in 15-month-old Plekhg5-deficient mice. **c** Body weight measurements of Plekhg5-deficient and control mice of different ages (2–4 months, *Plekhg5*$^{+/+}$: $n = 12$, *Plekhg5*$^{−/−}$: $n = 9$; 9–15 months, *Plekhg5*$^{+/+}$: $n = 19$, *Plekhg5*$^{−/−}$: $n = 42$; 18–25 months, *Plekhg5*$^{+/+}$: $n = 18$, *Plekhg5*$^{−/−}$: $n = 36$; two-way ANOVA; Bonferroni post-test). **d** Number of motoneurons was counted in lumbar spinal cord sections (3 months, $n = 5$; 12 months, $n = 5$; 24 months, $n = 3$; unpaired *t*-test; two-tailed). **e**, **f** Grip strength measurements of fore- **e** and hindlimbs **f**. **g** Rotarod performance (9–15 months, *Plekhg5*$^{+/+}$: $n = 9$, *Plekhg5*$^{−/−}$: $n = 10$; 18–24 months, *Plekhg5*$^{+/+}$: $n = 11$, *Plekhg5*$^{−/−}$: $n = 8$; two-way ANOVA; Bonferroni post-test). **h** Nissl-stained motoneurons in spinal sections from *Plekhg5*$^{+/+}$ and *Plekhg5*$^{−/−}$ mice. *Scale bar*: 40 μm. **i–l** Electrophysiological recordings of gastrocnemius and plantaris muscles (*Plekhg5*$^{+/+}$: $n = 6$, *Plekhg5*$^{−/−}$: $n = 6$; unpaired *t*-test; two-tailed). Latency **i** and amplitude **j** of muscle depolarization upon stimulation of the sciatic nerve. **k**, **l** Single motor unit potential (SMUP) **k** and motor unit number estimation (MUNE) **l** of the gastrocnemius muscle. **m** Succinate dehydrogenase (SDH) staining of the gastrocnemius muscle. *Scale bar*: 500 μm. **n** X-Gal staining of cross-sectioned spinal cord and gastrocnemius muscle. *Scale bar*: 20 μm. **o** Sciatic nerve conduction velocities (*Plekhg5*$^{+/+}$: $n = 6$, *Plekhg5*$^{−/−}$: $n = 6$; unpaired *t*-test; two-tailed) **p**, **q** Box- and whisker-plots show mechanical withdrawal thresholds **p** and heat withdrawal latencies (s) **q** of naive male *Plekhg5*$^{−/−}$ and control littermates. Mechanical withdrawal thresholds did not differ between genotypes **p**. *Plekhg5*$^{−/−}$ mice showed heat hypersensitivity as compared to control littermates (*Plekhg5*$^{+/+}$: $n = 5$, *Plekhg5*$^{−/−}$: $n = 5$; Mann–Whitney *U*-test)

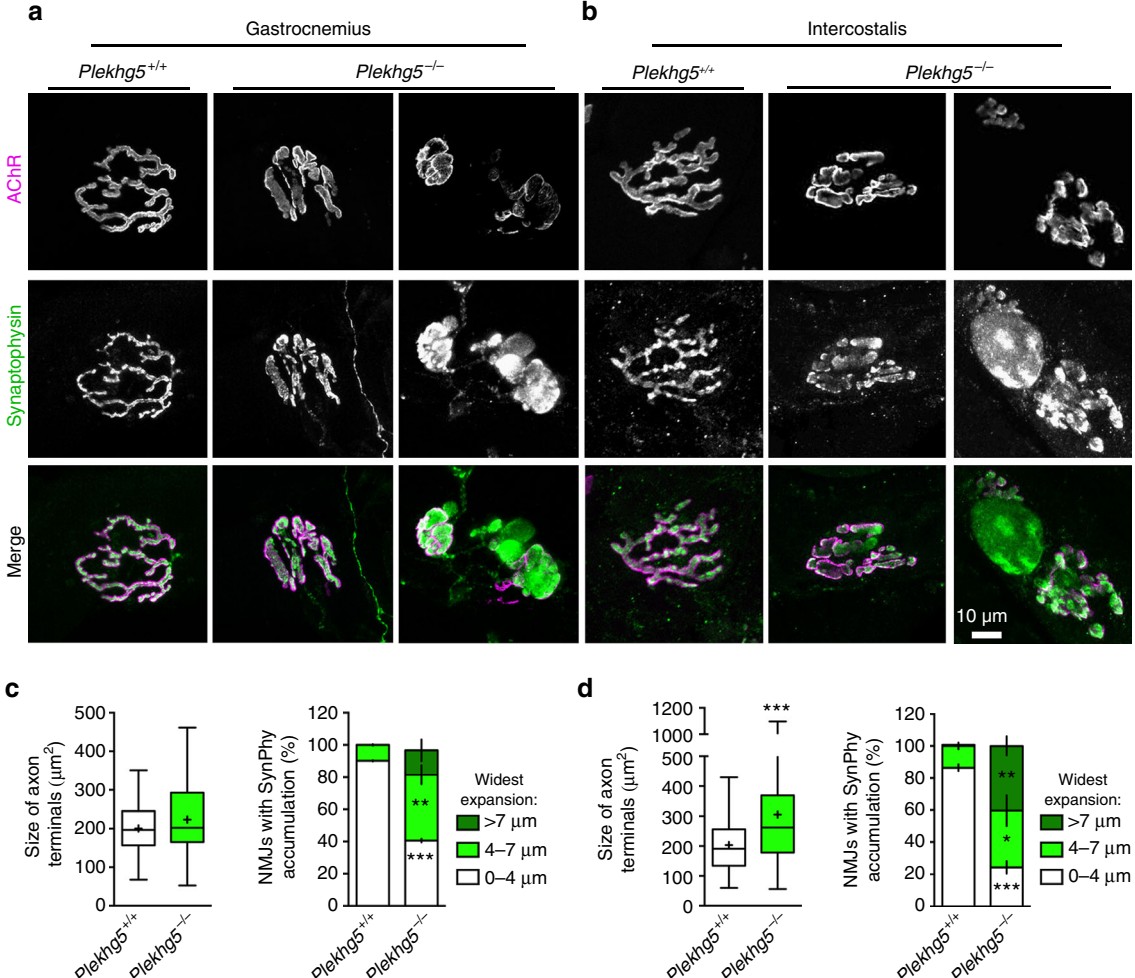

**Fig. 2** Degradation of neuromuscular junctions in Plekhg5-deficient mice. **a**, **b** NMJs within the gastrocnemius **a** and intercostal **b** muscles stained for BTX and synaptophysin. *Scale bar*: 10 μm. **c**, **d** Quantification of presynaptic area by evaluation of synaptophysin staining and measurement of the widest expansion of presynaptic sites (three animals per genotype with at least 15 NMJs analyzed per animal; mean ± SEM; two-way ANOVA; Bonferroni post-test)

terminals of $Plekhg5^{-/-}$ mice appeared swollen with accumulations of synaptophysin (Fig. 2a, c; Supplementary Fig. 3) and neurofilament-H (NF-H) (Supplementary Fig. 3).

Respiratory failure due to progressive impairment of the neuromuscular system has been reported as a main cause of death in ALS and the major trigger for premature death in SOD1 G93A mice[23]. We therefore investigated whether NMJs within intercostal muscles of Plekhg5-deficient mice also show signs of degeneration. We observed a phenotype similar as in the gastrocnemius muscle with deformed and swollen axon terminals (Fig. 2b, d), indicating that muscles necessary for respiration become denervated which could make a major contribution to premature death.

At the ultrastructural level, axon terminals of Plekhg5-deficient mice were filled with membrane fragments and cytoplasmic inclusions (Fig. 3a–e), similar as those observed in patients with ALS[24, 25]. Strikingly, synaptic vesicles in $Plekhg5^{-/-}$ motoneuron terminals were frequently enlarged, in contrast to synaptic vesicles in wild-type mice, which appeared smaller and of a more uniform size (Fig. 3f–h). Furthermore, the organization of actin filaments in axon terminals appeared altered, with more actin fibers in close neighborhood of synaptic vesicles in active zones (Fig. 3f, g). In line with these findings we detected elevated protein levels of several synaptic vesicle markers in sciatic nerve

lysates of $Plekhg5^{-/-}$ mice (Fig. 3i, j). The ultrastructural examination of NMJs also revealed axonal swellings in distal axons (Fig. 4a). Axons showed a highly disorganized cytoskeleton (Fig. 4b, c). Axonal swellings were also frequently detectable in spinal cord and sciatic nerve cross-sections (Fig. 4d–j). In contrast, we detected no axonal swelling in wild-type animals (Fig. 4j). Taken together, these data indicate that Plekhg5 deficiency results in degradation of axon terminals, but also affects axonal integrity in proximal parts of axons.

Next, we studied the impact of Plekhg5 deficiency on synapse-morphology in the central nervous system and stained brain sections for synaptophysin and Tuj1 (Supplementary Fig. 4). The overall morphology and layering of the hippocampus, cerebellum and cortex appeared normal (Supplementary Fig. 4). In contrast to the abnormally swollen appearance of motoneuron terminals with synaptophysin-accumulations, synapses within the brain appeared unaffected in Plekhg5-deficient mice (Supplementary Fig. 4). These observations are in line with our data on normal cognitive function in Plekhg5-deficient mice.

**Plekhg5 regulates autophagy in motoneurons.** Under physiological conditions damaged organelles are removed from axon terminals by autophagy[1, 2, 6]. To assess whether

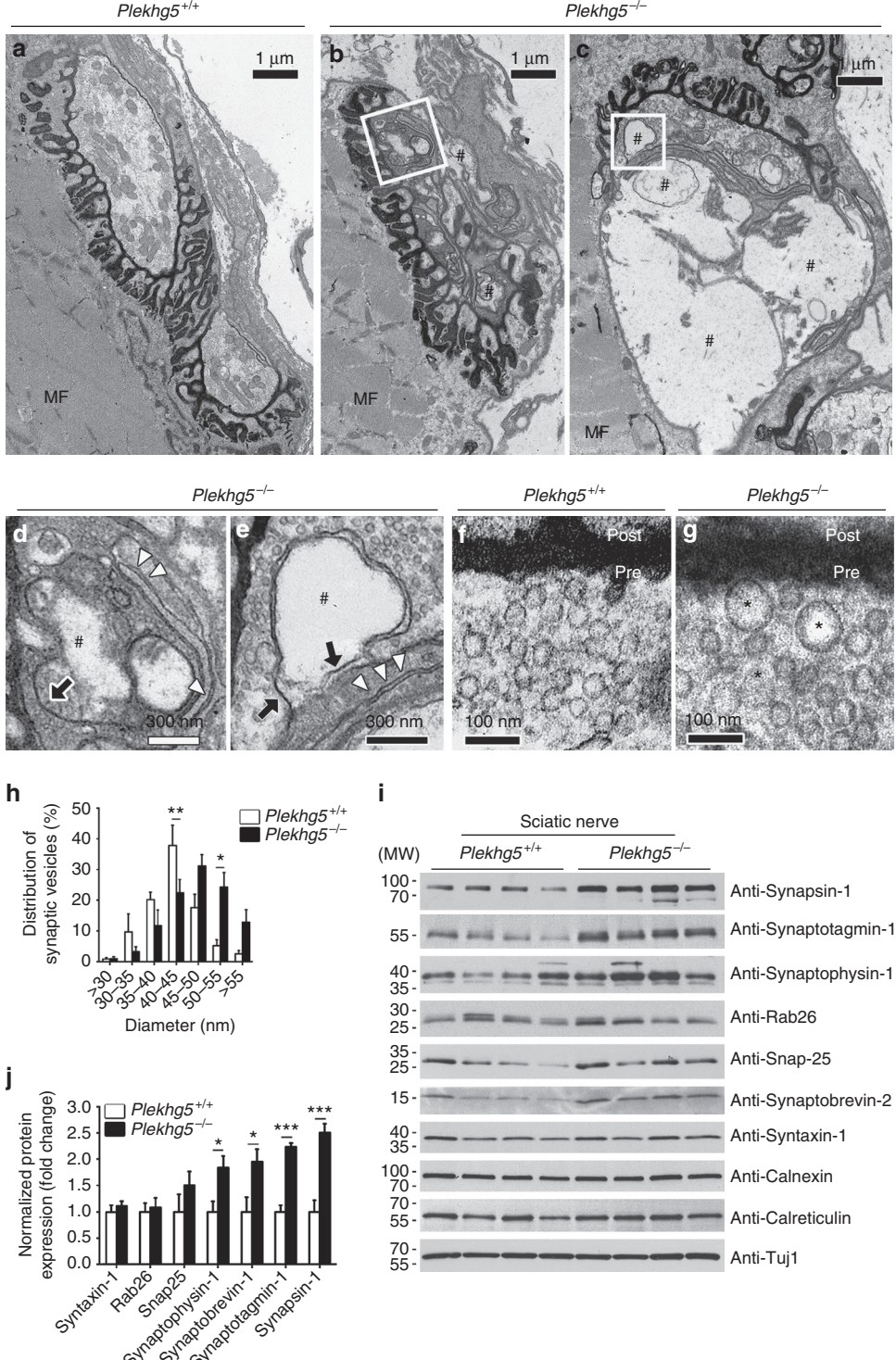

**Fig. 3** Structural alterations and accumulation of synaptic vesicle proteins in Plekhg5-deficient mice. **a–c** Neuromuscular junction of *Plekhg5+/+* **a** and *Plekhg5−/−* **b**, **c** mice. # labels empty inclusions. *MF* myofibers. *Scale bar*: 1 μm. **d** Membrane fragments in nerve terminals of Plekhg5-deficient mice. *Scale bar*: 300 nm. **e** Double membrane fragment forming an inclusion. Arrow points to single membrane. Arrowheads point to double membrane structures. # labels inclusions. *Scale bar*: 300 nm. **f**, **g** Synaptic vesicles in Plekhg5-deficient mice **g** appear frequently enlarged, in contrast to synaptic vesicles in wild-type mice **f**, which appear smaller and more uniform in size. Asterisks label enlarged synaptic vesicles. *Scale bar*: 100 nm. **h** Quantification of synaptic vesicle diameter. (Synaptic vesicles of five NMJs were analyzed per genotype. Mean ± SEM; two-way ANOVA; Bonferroni post-test). **i** Expression of several synaptic vesicle markers in sciatic nerve lysates of four mice per genotype. **j** Quantification of western blot analysis. Mean ± SEM; two-way ANOVA; Bonferroni post-test). Images have been cropped for presentation. Full size images are presented in Supplementary Fig. 7

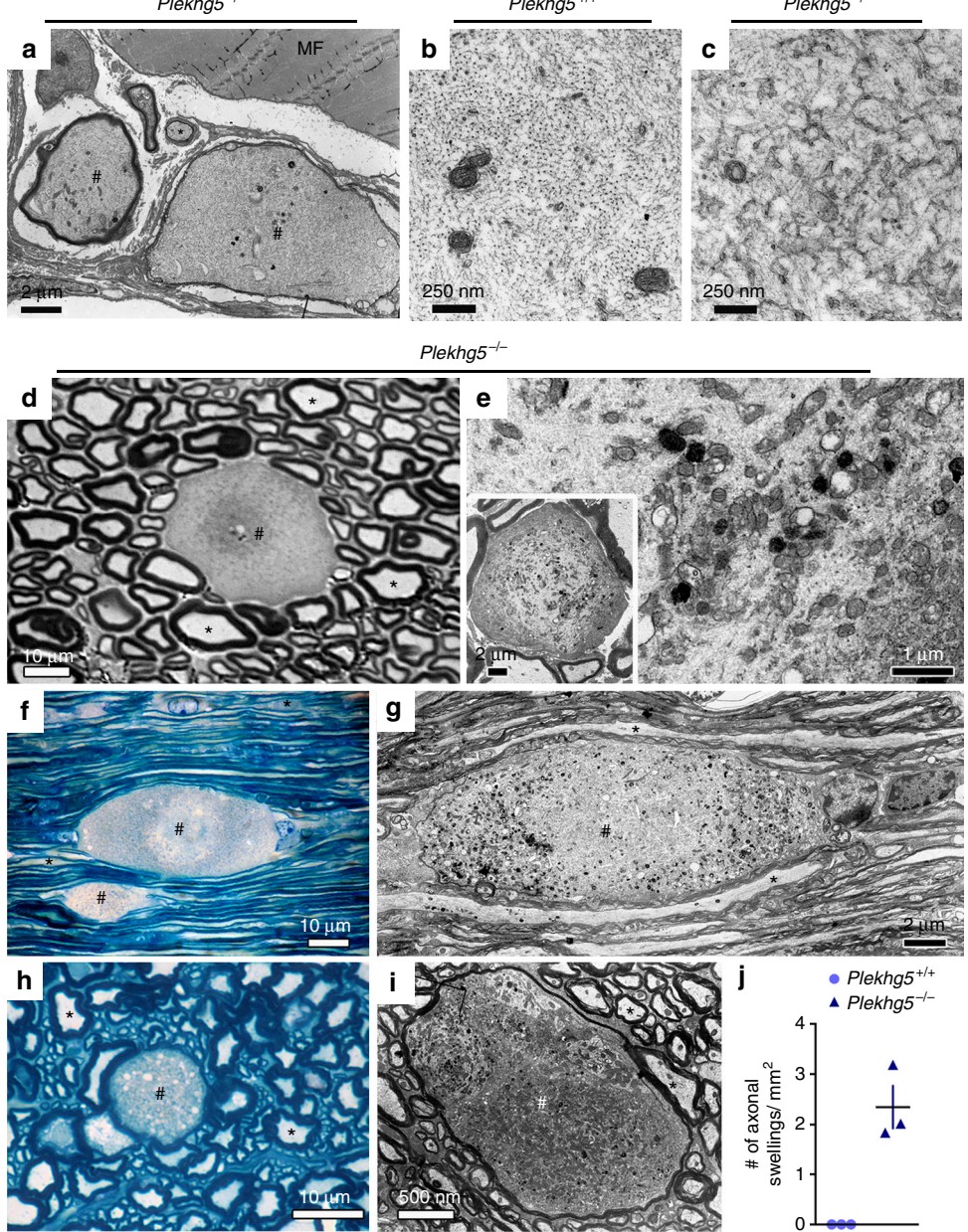

**Fig. 4** Loss of Plekhg5 impairs axonal integrity. **a** Swellings in distal motor axons. MF, muscle fiber. *Scale bar*: 2 μm. **b**, **c** High magnification micrographs of axons from wild-type **b** and Plekhg5-deficient **c** mice with altered cytoskeleton organization in *Plekhg5^(−/−)* mice. *Scale bar*: 250 nm. **d** Semi-thin cross-section of sciatic nerve from Plekhg5-deficient mice showing an axonal swelling. *Scale bar*: 10 μm. **e** High magnification micrograph of an axonal swelling from sciatic nerve. *Scale bar*: 1 μm. *Inset, Scale bar*: 2 μm. **f**–**i** Longitudinal- **f**, **g** and cross- **h**, **i** sections of lumbar spinal cord showing axonal swellings within the white matter of *Plekhg5^(−/−)* mice. **f**, **h** *Scale bar*: 10 μm. **g**, **i** Fine structure of axonal swellings in spinal cord white matter of Plekhg5-deficient mice. # labels axon swellings; asterisks label axons with unaltered morphology. **g** *Scale bar*: 2 μm. **h** *Scale bar*: 10 μm. **i** *Scale bar*: 500 nm. **j** Quantification of axonal swellings in spinal cord semi-thin cross-sections. Three animals per genotype were analyzed. Each *data point* represents the mean of 10 sections from individual animals with a distance of 100 μm between each section

dysregulated autophagy causes the accumulation of synaptic vesicles at nerve terminals we examined the autophagic flux in Plekhg5-deficient motoneurons (Fig. 5). Since autophagosomes rapidly fuse with lysosomes, changes in the biogenesis of autophagosomes are hardly detectable under basal conditions[26, 27]. Therefore, cultured motoneurons were transduced with lentiviruses expressing GFP-RFP-LC3 and treated with bafilomycin A1 after 7 days in culture (Fig. 5a–c). Bafilomycin A1 blocks the fusion of autophagosomes and lysosomes resulting in an enrichment of LC3-II positive autophagosomes. Since fusion of autophagosomes and lysosomes quenches the fluorescence of

GFP, RFP positive structures selectively mark autolysosomes, whereas autophagosomes appear positive for both RFP and GFP[27]. Under basal conditions we detected no difference in the number of autophagosomes and autolysosomes in the soma of wild-type and *Plekhg5^(−/−)* cells. However, upon treatment with bafilomycin A1 Plekhg5-deficient cells displayed a significantly reduced number of autophagosomes in comparison to wild-type motoneuron somata (Fig. 5a, b). In axons of *Plekhg5^(−/−)* motoneurons, a reduced number of autophagosomes was detected in both untreated cells and bafilomycin A1 treated cells (Fig. 5a, c). In agreement with previous reports we only identified very few

autolysosomes in axons of cultured motoneurons, confirming that full maturation of autophagosomes to lysosomes mostly occurs within or close to the soma[1, 26]. The number of autolysosomes was not altered indicating that acidification of autolysosomes, as a late step of maturation, was unaffected (Fig. 5a–c). To examine whether Plekhg5 deficiency causes impaired axonal transport of autophagosomes, motoneurons were transduced with RFP-GFP-LC3, and the movement of GFP-LC3+ structures was analyzed (Fig. 5d–f). Plekhg5−/− motoneurons showed a reduced number of autophagosomes

moving retrogradely towards the soma (Fig. 5f), whereas the relative number of retrogradely moving autophagosomes appeared unaltered (Fig. 5e; Supplementary Movie 1, 2). Reduced protein levels of endogenous LC3-II were also biochemically detected in Plekhg5-deficient neurosphere-derived neurons upon treatment with bafilomycin A1 by western blotting (Fig. 5g, h). This reduction of LC3-II levels was also apparent in spinal cord extracts of 24-month-old mice (Fig. 5i, j). Taken together, these data emphasize that the biogenesis of autophagosomes is impaired in Plekhg5-deficient motoneurons

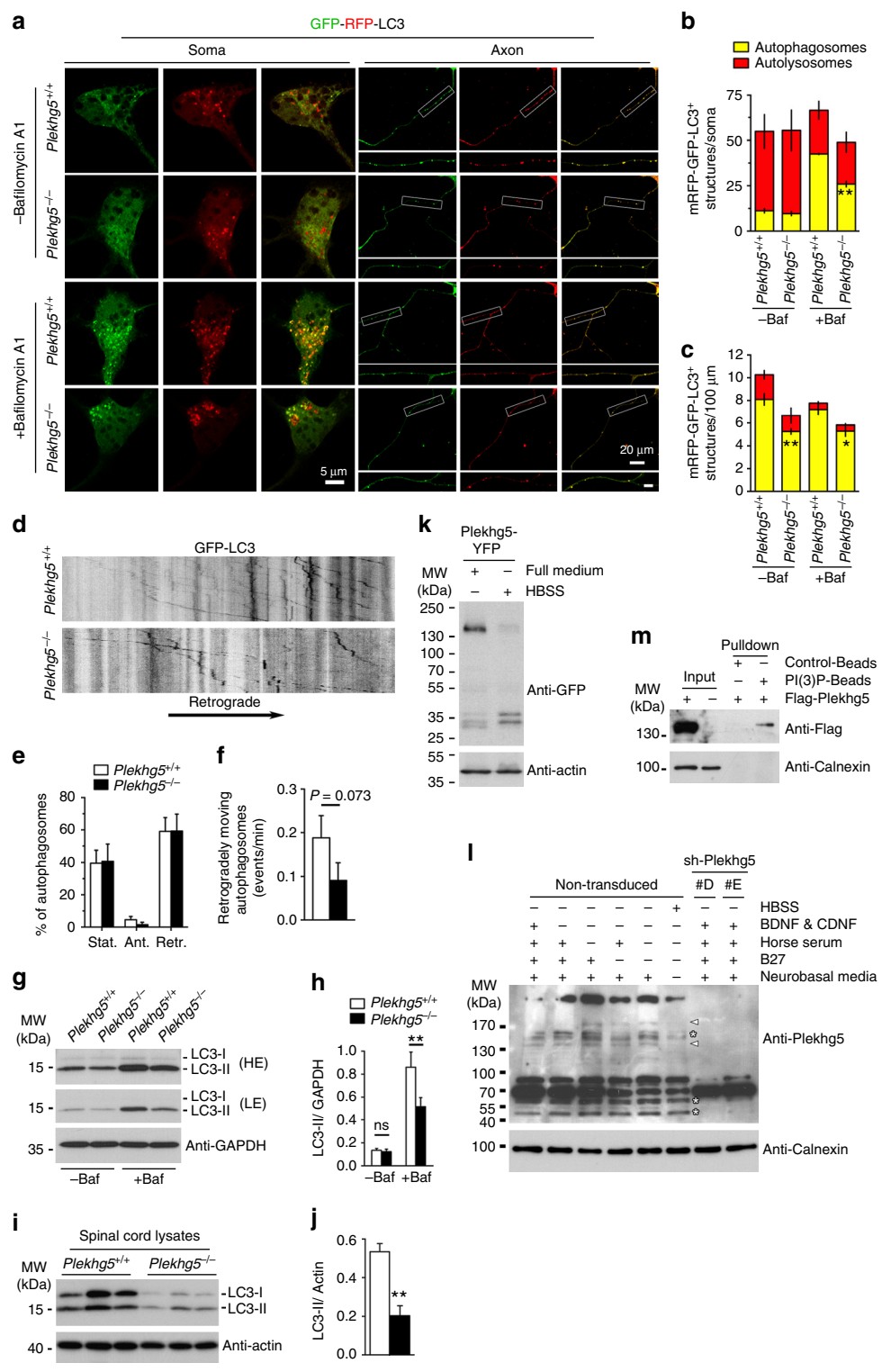

resulting in a reduced number of retrogradely transported autophagosomes. The remaining autophagosomes are transported normally.

An essential process for autophagosome formation is the production of phosphatidylinositol 3-phosphate (PI(3)P) by phosphatidylinositol-3-kinase class III at preautophagosomal membranes[28]. Pleckstrin homology (PH) domains form a conserved family that mediates protein association to inositol lipids like PI(3)P. We analyzed whether Plekhg5 binds to PI(3)P using extracts from mouse motoneuron-like hybrid cells (NSC34) expressing Flag-tagged Plekhg5. We could precipitate Plekhg5 with PI(3)P-coated beads but not with control beads (Fig. 5m) suggesting that Plekhg5 is able to associate with PI(3)P at preautophagosomal membranes[28]. For monitoring the fate of Plekhg5 upon induction of autophagy, Plekhg5-YFP transfected NSC34 cells were starved for 4 h in HBSS and the expression of Plekhg5-YFP was analyzed by western blot (Fig. 5k). In contrast to a marked reduction of full-length Plekhg5-YFP, the signal for smaller protease-resistant YFP-positive fragments increased (Fig. 5k). Nutrient deprivation differently affects cell lines and primary neurons. Therefore, we also performed this experiment with cultured motoneurons by replacing enriched motoneuron-media with different media each deprived of specific nutrient components (Fig. 5l). As observed in NSC34 cells, nutrient deprivation resulted in degradation of Plekhg5. In contrast to overexpressed Flag-Plekhg5, endogenous Plekhg5 appeared as multiple bands that were differently affected by nutrient deprivation (Fig. 5l). The identity of the bands that were detected by western blot analysis was controlled by sh-RNA-mediated knockdown of Plekhg5 (Fig. 5l; Supplementary Fig. 5a, b). In summary, these experiments suggest that Plekhg5 is able to bind PI(3)P, placing Plekhg5 to the inner autophagosomal membrane. This idea is further supported by the degradation of Plekhg5 upon autophagy induction. Fusion of autophagosomes with lysosomes results in digestion of proteins in the lumen of the autophagosome including proteins localized to the inner autophagosomal membrane[29].

Next, we determined whether impaired autophagy affects survival and/or morphology of Plekhg5-depleted motoneurons in vitro. Plekhg5-depleted motoneurons grew shorter axons, whereas cell survival and dendritic complexity was not affected (Fig. 6a, b; Supplementary Fig. 5). In order to find out how altered autophagy leads to axonal growth defects, we characterized the morphology of axon terminals in $Plekhg5^{-/-}$ motoneurons. Growth cones of cultured $Plekhg5^{-/-}$ motoneurons appeared atrophic with a significantly smaller size and with dysmorphic densely packed accumulation and clustering of synaptophysin-positive synaptic vesicles leading to a reduced number of F-actin filaments (Fig. 6c, d). We next looked at the expression pattern

of Rab26 in axons of cultured motoneurons. Rab26 is a small GTPase enriched on synaptic vesicles designated for delivery to preautophagosomal structures[15]. In its GTP-bound form Rab26 is an effector of Atg16L leading to recruitment of the autophagy machinery to synaptic vesicles. In Plekhg5-deficient cells Rab26-positive structures accumulate in regions directly at active zones, in contrast to wild-type motoneurons where these structures are found in more proximal areas of axonal growth cones (Fig. 6c, d).

To examine these morphological alterations in more depth, we performed SIM microscopy (Fig. 6e). These high-resolution images confirmed the data obtained by confocal microscopy providing additional evidence that the number of Rab26-positive vesicles increased, and the actin cytoskeleton disorganized (Fig. 6e). These findings support the in vivo structural analyses demonstrating marked alterations in cytoskeletal structures (Fig. 4b, c), which might cause the defect in axon growth.

**Plekhg5 functions as a GEF for Rab26.** Activation of GTPases requires dissociation of protein-bound GDP, an intrinsically slow process that is accelerated by specific GEFs[30]. To explore whether Plekhg5 regulates the activity of Rab26, neurosphere-derived cortical neurons were transduced with EGFP-Rab26 and labeled with $^{32}P$ for 4 h. Subsequently, EGFP-Rab26 was immunoprecipitated and GTP- or GDP-bound Rab26 was separated by thin layer chromatography (TLC) (Fig. 7a), revealing a significantly reduced GTP/GDP ratio (Fig. 7b). Next, we asked whether Plekhg5 is able to directly act as a GEF for Rab26 in a cell-free system (Fig. 7c–h). Using homogenously purified recombinant proteins we analyzed the ability of Plekhg5 to catalyze GDP dissociation of Rab26 and observed a marked acceleration of GTP exchange in the presence of the DH-PH (Dbl-homologous-Pleckstrin homology) tandem domain of Plekhg5 for Rab26 (Fig. 7g, h), but not for Rab5, Rab27b, or Rab33b (Fig. 7d–f). These data demonstrate that Plekhg5 acts as a GEF for Rab26.

We then followed this line of experiments and tested wild-type and a constitutively active form of Rab26, Rab26Q123L (Rab26-QL), in cultured motoneurons (Fig. 8). As previously described, wild-type EGFP-Rab26 accumulated in vesicles within axons[15]. In Plekhg5-deficien motoneurons, the size of these vesicles was significantly reduced (Fig. 8a, c), and the number of EGFP-Rab26 vesicles was also modestly reduced (Fig. 8a, d). With constitutive active EGFP-Rab26, the number and size of these vesicles normalized and no differences between wild-type and Plekhg5-deficient cells were detectable (Fig. 8a, c, d). In correlation with these findings, EGFP-Rab26-QL also fully restored axon growth

**Fig. 5** Plekhg5 regulates biogenesis of autophagosomes. **a** Motoneurons of $Plekhg5^{+/+}$ and $Plekhg5^{-/-}$ mice were transduced with mRFP-GFP-LC3-expressing lentiviruses and cultured for seven days. At day seven cells were treated with Bafilomycin A1 for four hours or left untreated and the number of autophagosomes (mRFP$^+$-GFP$^+$-LC3) and autolysosomes (mRFP$^+$-GFP$^-$-LC3) was determined in the soma **b** and axon **c** (three independent experiments with 15 cells analyzed in each experiment; mean ± SEM; two-way ANOVA; Bonferroni post-test). Soma, scale bar: 5 μm. Axon, scale bar: 20 μm. Axon (blow up), scale bar: 5 μm. **d** Representative kymographs of autophagosome motility. Cultured motoneurons were transduced with mRFP-GFP-LC3 and the movement of GFP positive punctae in axons was monitored for 20 min. **e** Proportion of retrogradely, anterogradely or stationary/bidirectionally moving autophagosomes (mean ± SEM; n = 10 cells; two-way ANOVA; Bonferroni post-test). **f** Number of retrogradely moving autophagosomes per minute (mean ± SEM; n = 10 cells; Student's t-test; one-tailed). **g** Western blot analysis of LC3 expression in neurosphere-derived cortical neurons from control and Plekhg5-deficient mice. Cells were treated with 400 nM Bafilomycin A1 for 4 h or left untreated. LE low exposure, HE high exposure. **h** Quantification of LC3 western blots. (Mean ± SEM; n = 3; mean ± SEM; one-way ANOVA). **i** Expression of LC3 in spinal cord lysates of three mice per genotype. **j** Quantification of LC3 Western blots from spinal cord extracts. (mean ± SEM; n = 4; mean ± SEM; Student's t-test; two-tailed). **k** NSC34 cells were transfected with Plekhg5-YFP. Seventy-two hours after transfection, cells were starved for 4 h in HBSS or left untreated. The expression of Plekhg5-YFP was analyzed by western blot. **l** Motoneurons were cultured for seven days in motoneuron-media. At day seven cells were cultured for 4 h with different media each deprived of specific components. Subsequently, motoneurons were lysed and the expression of endogenous Plekhg5 was analyzed by western blot. Asterisks label bands that do not change upon nutrient deprivation. Arrowheads point to bands that change upon nutrient deprivation. **m** NSC34 cells were transfected with Flag-Plekhg5. Seventy-two hours after transfection, cells were lysed and Flag-Plekhg5 was pulled-down with PI(3)P-coated beads. Western blot images have been cropped for presentation. Full size images are presented in Supplementary Fig. 7

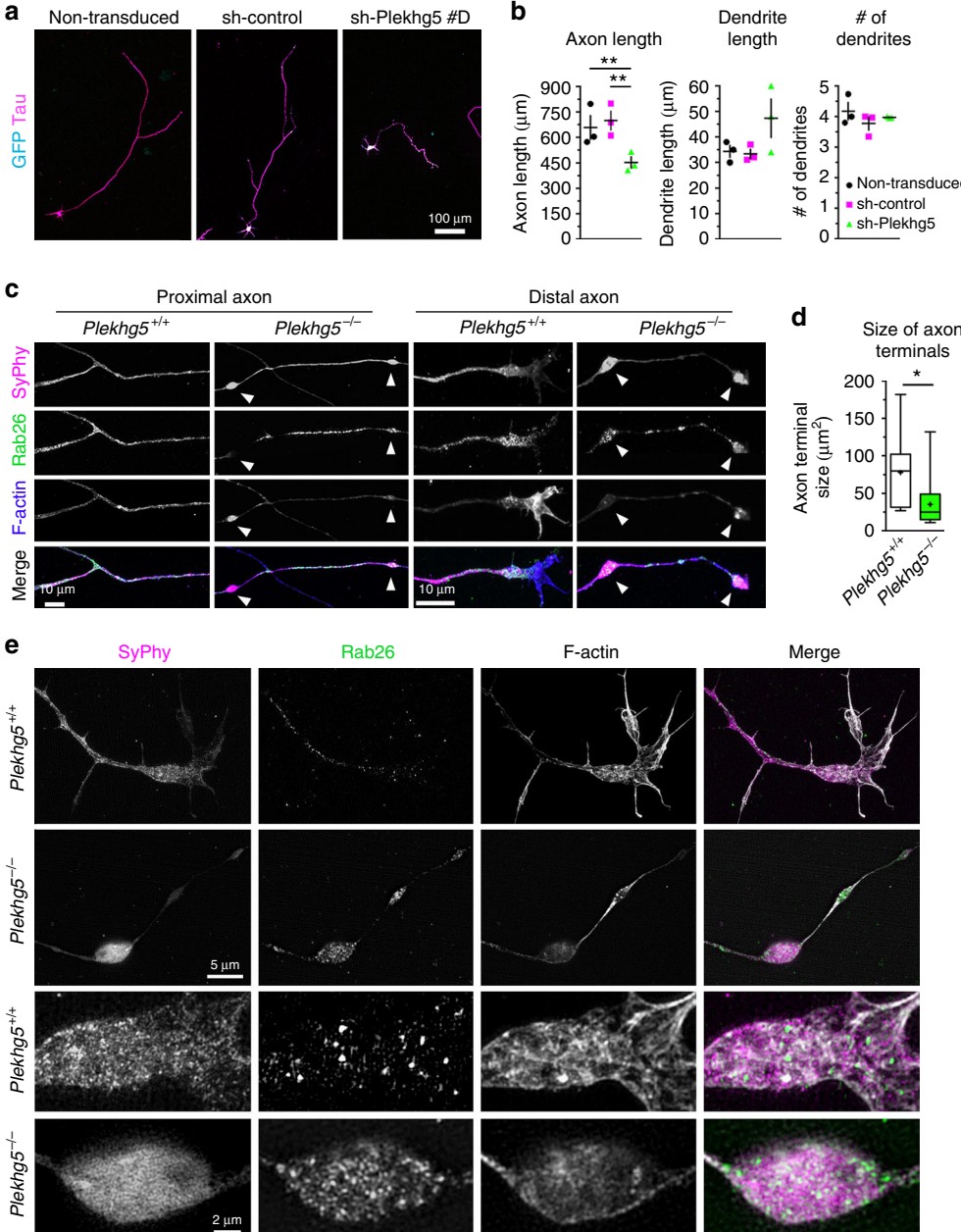

**Fig. 6** Depletion of Plekhg5 results in axon growth defects and degeneration of axon terminals in vitro. **a** Motoneurons were transduced with sh-control or sh-Plekhg5 #D lentiviruses or left untreated. *Scale bar*: 100 μm. **b** Seven days after sh-RNA transduction, knockdown of Plekhg5 reduced axon length, whereas dendrite length and number of dendrites were not affected. Each data point represents the mean of one individual experiment with at least 20 cells analyzed. One-way ANOVA, Bonferroni post-test. **c–e** Motoneurons were cultured for seven days and stained for synaptophysin, Rab26 and F-actin and imaged by confocal **c** and SIM microscopy **e**. **c** *Scale bar*: 10 μm. **d** Quantification of axon terminal size ($n = 15$ cells; unpaired *t*-test; two-tailed). **e** Motoneurons were cultured for 7 days and stained for synaptophysin, Rab26 and F-actin and imaged by SIM microscopy. Axons of Plekhg5$^{-/-}$ motoneurons display synaptophysin positive swellings. *Scale bar*: 10 μm. *Scale bar* (*blow up*): 2 μm

defects of Plekhg5-deficient motoneurons (Fig. 8b, e). In contrast, EGFP-Rab26-QL only moderately enhanced axon growth in wild-type cells (Fig. 8b, e).

In order to study the effects of constitutive Rab26 on axonal autophagy, we constructed lentiviral vectors that simultaneously express RFP-GFP-LC3 and Flag-Rab26-WT or Flag-Rab26-QL (Fig. 8f, g), respectively. We then quantified the number of RFP-GFP-LC3 positive structures in proximal axons and found that the number of autophagosomes was reduced in Plekhg5-deficient motoneurons (Fig. 8h, i). This reduction did not occur in Plekhg5$^{-/-}$ motoneurons expressing constitutive active Rab-QL (Fig. 8h, i). In agreement with the data shown in

Fig. 5a–c, the number of autolysosomes remained unaltered. These data confirm the conclusion that Plekhg5 acts as a GEF for Rab26 and contributes to autophagosome biogenesis.

**Plekhg5 modifies ER-stress in SOD1 G93A motoneurons.** Protein misfolding and endoplasmatic reticulum (ER) stress is a common disease mechanism in different models of motoneuron disease[31–36]. Deficiency of global autophagy in neurons results in an impairment of proteostasis characterized by accumulation of polyubiquitinated proteins in inclusion bodies[8, 9]. Plekhg5-deficient mice did not show any accumulation of

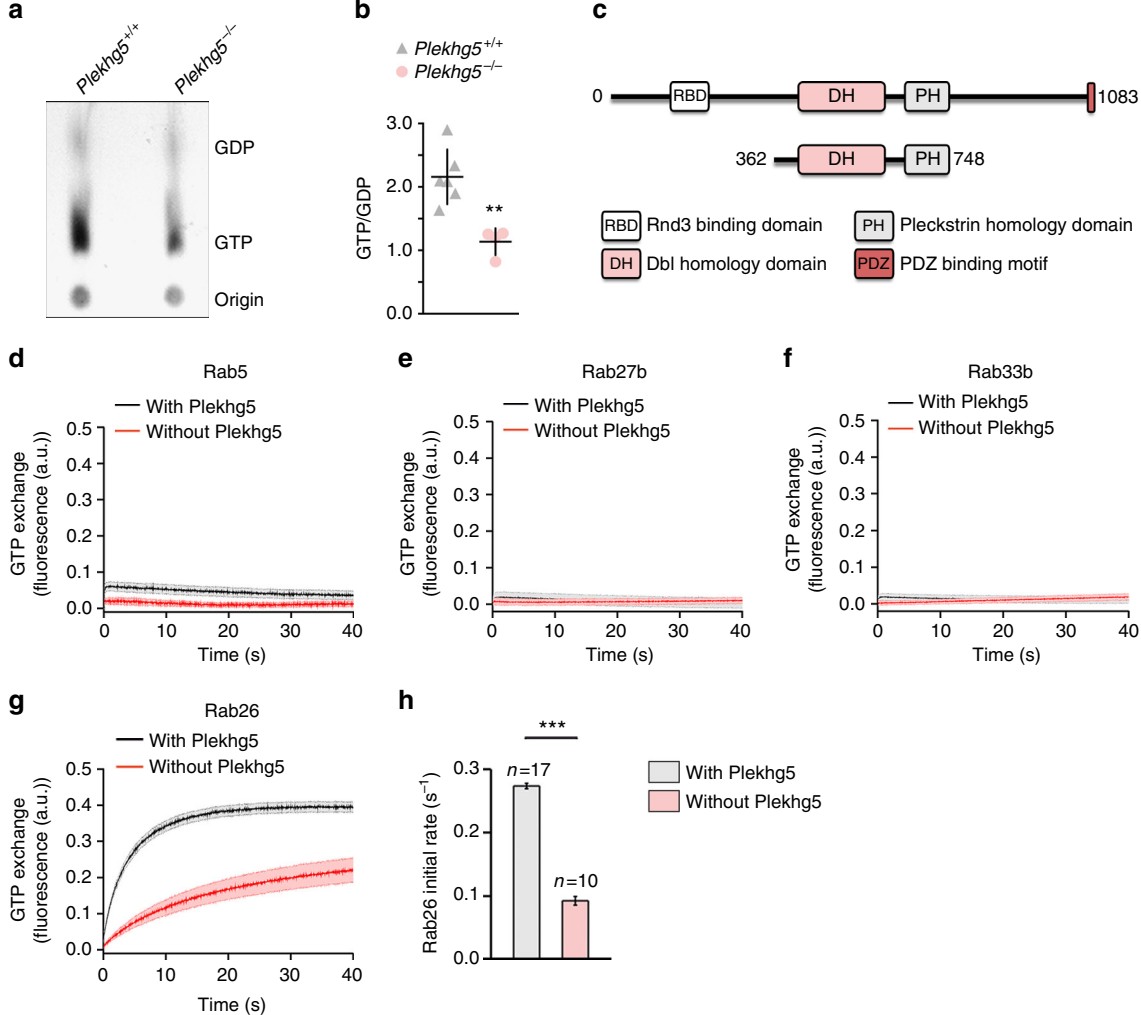

**Fig. 7** Plekhg5 functions as a GEF for Rab26. **a** Neurosphere-derived cortical neurons were transduced with EGFP-Rab26 and labeled with P[32] for 4 h. EGFP-Rab26 was immunoprecipitated and GTP- or GDP-bound Rab26 was separated by TLC. **b** Densitometric quantification of GTP/GDP ratios (each *data point* represents one independent experiment. Mean ± SEM; unpaired *t*-test; two-tailed). **c–h** Exchange activity of Plekhg5 on different GTPases. **c** Scheme of the DH-PH tandem domain, which was purified and used for the biochemical assays. Exchange activity of Plekhg5 on Rab5 **d**, Rab27b **e**, Rab33b **f**, and Rab26 **g** was measured by monitoring the fluorescent increase of Mant-GppNHp upon binding to Rab proteins. **g** The initial GTP exchange rate of Rab26 was evaluated using the first 10 s of each time course. (Mean ± SEM; Student's *t*-test; two-tailed)

polyubiquitinated proteins at axon terminals or in motoneuron cell bodies as shown by immunohistochemical staining (Supplementary Fig. 6a, b). In addition, we were not able to detect any differences in the levels of polyubiquitinated proteins by western blotting (Supplementary Fig. 6c). In contrast, the levels of Chop and IRE1α were markedly elevated in *Plekhg5*[−/−] spinal cord extracts (Fig. 9c, d). Thus, the unfolded protein response (UPR) seems to be activated in Plekhg5-deficient mice. We also tested the expression levels of heat shock protein family members HSP70 and HSP90 and the ER chaperone Calreticulin by western blot analyses and found significantly decreased expression of all three chaperones (Fig. 9a, b). Together, these data suggest that Plekhg5 deficiency results in ER-stress without affecting the turnover of polyubiquitinated proteins.

In order to investigate the relevance of impaired synaptic vesicle autophagy for motoneuron vulnerability in a well-characterized model of ALS we depleted Plekhg5 in SOD1 G93A motoneurons in vitro (Fig. 9e–g). We analyzed the effect of Plekhg5 knockdown with two independent sh-RNA lentiviral constructs on embryonic SOD1 G93A motoneurons and investigated ER-stress as a pathogenic mechanism. ER-stress

precedes disease onset in SOD1 transgenic mice[37] and modulation of the UPR has been shown to modulate disease onset and progression in SOD1 transgenic mice[38, 39]. PERK phosphorylation, BIP expression and Chop activation were enhanced in a supra-additive manner in SOD G93A motoneurons with additional Plekhg5 depletion (Fig. 9e, f). Upon knockdown of Plekhg5, SOD1 G93A motoneurons also showed a decrease in survival (Fig. 9g). Interestingly, this effect was observed in embryonic motoneurons representing a developmental stage long before disease becomes apparent in vivo. We did not observe any significant changes in the levels of endogenous and transgenic SOD1 when Plekhg5 is depleted (Fig. 9e, f). Therefore, Plekhg5 is not involved in the degradation of the mutant SOD1 protein. These data indicate that both disease mechanisms have additive effects, cumulating in an elevated load of ER-stress.

## Discussion
In summary, our data show that Plekhg5 functions as a GEF for Rab26, and thus regulates autophagy of synaptic vesicles in axon terminal of motoneurons. Disruption of this process leads

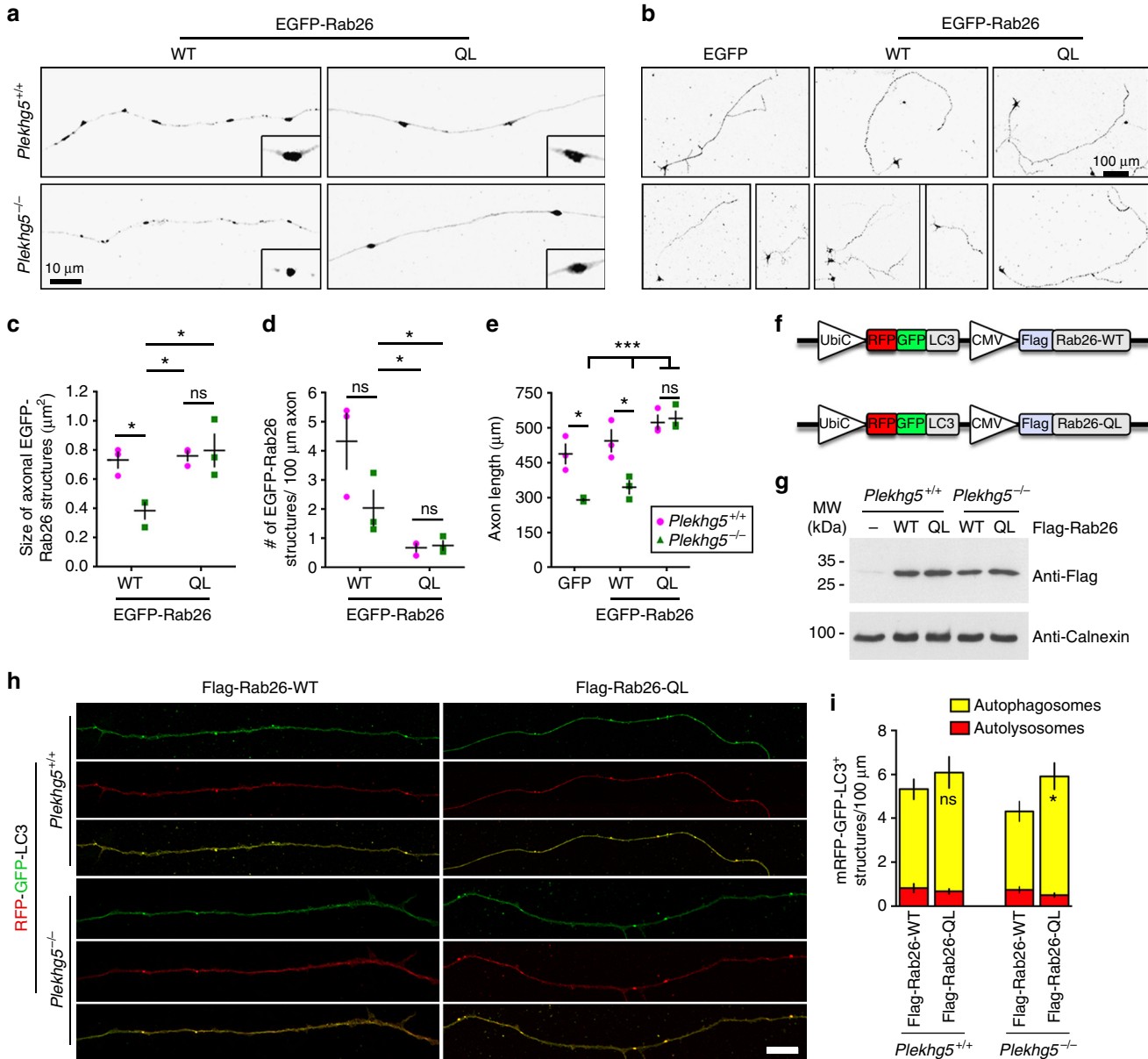

**Fig. 8** Expression of constitutively active Rab26 rescues axonal growth and autophagy defects in Plekhg5-deficient cells. **a** Representative images of GFP-Rab26-WT or GFP-Rab26-QL positive structures in axons of control cells and Plekhg5-deficient cells. *Scale bar*: 10 μm. **b** Morphology of GFP, GFP-Rab26-WT or GFP-Rab26-QL expressing motoneurons cultured for 7 days. *Scale bar*: 100 μm. **c**, **d** Size and number of axonal EGFP-Rab26 structures. **e** Axon length of GFP, GFP-Rab26-WT, or GFP-Rab26-QL expressing motoneurons isolated from *Plekhg5+/+* and *Plekhg5−/−* mice (each *data point* represents one individual experiment with 15 cells analyzed in each experiment; mean ± SEM; two-way ANOVA). **f** Scheme of lentiviral vectors for simultaneous expression of RFP-GFP-LC3 and Flag-Rab26-WT or Flag-Rab26-QL, respectively. **g** Western blot analysis of Flag-Rab26-WT and Flag-Rab26-QL expression. Images have been cropped for presentation. Full size images are presented in Supplementary Fig. 7. **h** Motoneurons of *Plekhg5+/+* and *Plekhg5−/−* mice expressing mRFP-GFP-LC3 and Flag-Rab26-WT or Flag-Rab26-QL were cultured for 7 days and the number of mRFP-GFP-LC3 positive structures was analyzed. *Scale bar*: 10 μm. **i** Number of autophagosomes and autolysosomes upon expression of Flag-Rab26-WT and Flag-Rab26-QL (three independent experiments with 10 cells analyzed in each experiment; mean ± SEM; two-way ANOVA; Bonferroni post-test)

to enlarged synaptic vesicles and accumulation of synaptic proteins resulting in dysfunction of neuromuscular junctions and motoneuron disease.

Our data suggest that Plekhg5 associates with PIP(3) on preautophagosomal membranes and regulates the activity of Rab26 by exchanging GDP to GTP. The observation that Plekhg5 does not regulate Rab5, Rab27b, or Rab33b activity suggests relative specificity, but does not exclude the possibility that Plekhg5 could also regulate the activity of other small GTPases. However, the observation that constitutively active Rab26 rescues

the phenotype in Plekhg5-deficient motoneurons suggests that the GEF activity for Rab26 is central for altered autophagy of synaptic vesicles in motoneurons. In its GTP-bound form Rab26 recruits Atg16l to synaptic vesicles for initiating the delivery of synaptic vesicles to autophagosomes. Two previously identified interaction partners of Plekhg5, Synectin[13], and the 14-3-3γ-protein[40], are involved in autophagy regulation[41, 42]. Both proteins have been described to interact with PI(3)P-kinase, resulting in the inhibition of autophagy biogenesis. The 14-3-3γ-protein acts as a scaffold that inhibits the activity of PI(3)P-kinase. In the

same manner, the 14-3-3γ-protein inhibits the GEF activity of Plekhg5. Furthermore, protein kinase D mediated phosphorylation of Ser92 at the N-terminus of Plekhg5 prevents the interaction between these proteins[40]. It is tempting to speculate that phosphorylation of Plekhg5 is the upstream signal to release 14-3-3γ protein from the complex with Plekhg5 and probably also from PI(3)P-kinase. This might result in initiation of autophagy and activation of Rab26.

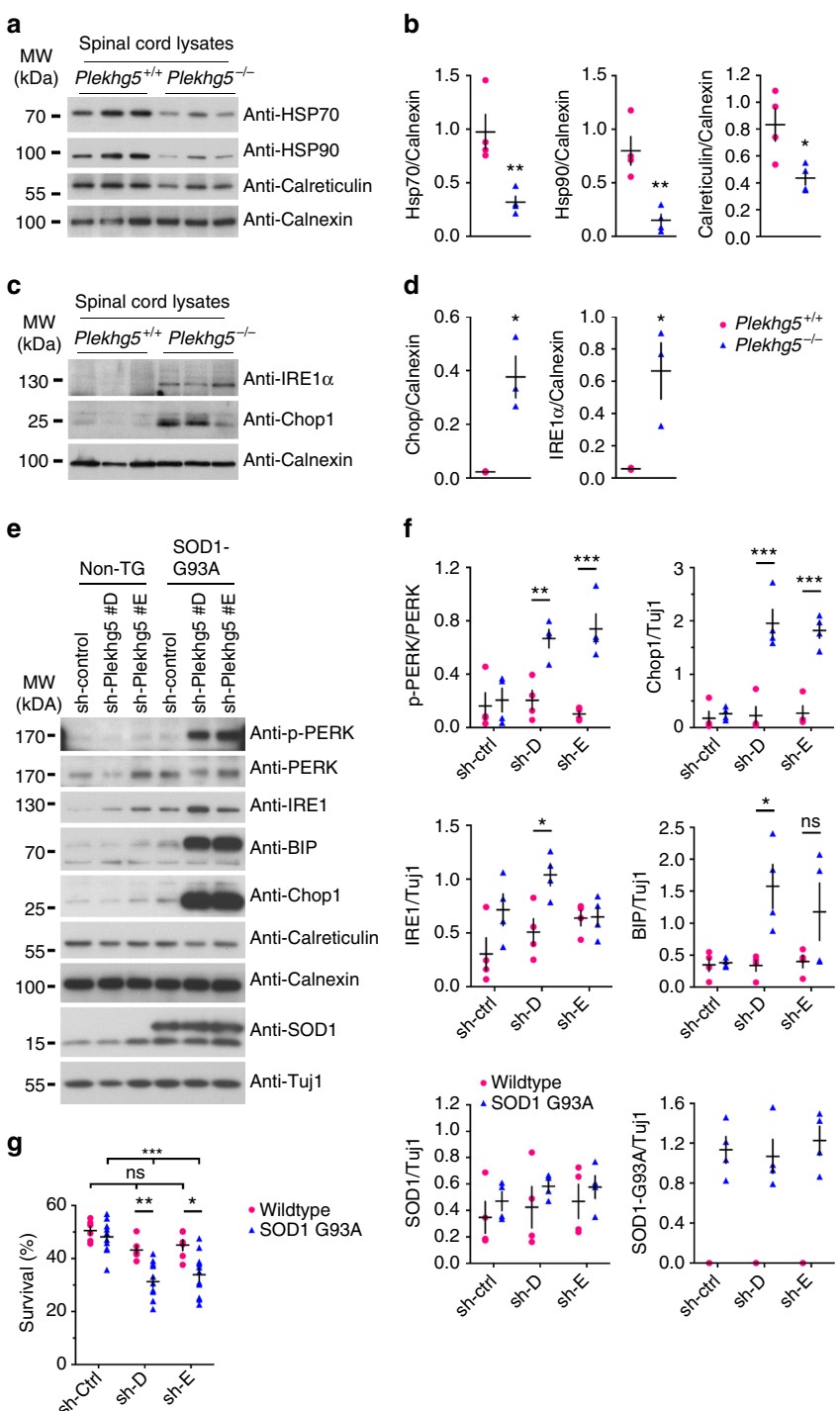

**Fig. 9** Plekhg5 depletion in SOD1 G93A motoneurons results in elevated ER-stress. **a** Expression of HSP70, HSP90, Calreticulin, and Calnexin in spinal cord lysates from three animals per genotype. **b** Quantification of western blot shown in **a** (each *data point* represents expression levels of one animal; unpaired *t*-test; two-tailed). **c** Expression of IRE1α and Chop1 in spinal cord lysates from three animals per genotype. **d** Quantification of western blot shown in **c** (each *data point* represents expression levels of one animal; unpaired *t*-test; two-tailed). **e** SOD1 G93A and non-transgenic motoneurons were depleted of Plekhg5 and several ER-stress markers were examined after 7 days in culture. **f** Quantification of western blots shown in **e** (each *data point* represents one individual experiment; mean ± SEM; unpaired *t*-test; two-tailed). **g** Survival of SOD1 G93A motoneurons decreased upon knockdown of Plekhg5 using two independent sh-RNA constructs (each *data point* represents the % of motoneuron-survival from one individual embryo. At least 50 motoneurons were evaluated from one embryo; mean ± SEM; two-way ANOVA; Bonferroni post-test). Images have been cropped for presentation. Full size images are presented in Supplementary Fig. 7

Deficiency of global autophagy in neurons results in an impairment of proteostasis characterized by accumulation of polyubiquitinated proteins in inclusion bodies[8, 9]. Our data suggest that Plekhg5 deficiency does not impair global protein homeostasis indicating that the defect in *Plekhg5*[−/−] mice is restricted to synaptic vesicles and does not involve misfolding or aggregation of cytosolic proteins. Thus, *Plekhg5*[−/−] mice represent a model of motoneuron disease without poly-ubiquitinated protein inclusion in which defective autophagy of synaptic vesicles stands in the center of the pathophysiology. Such cases of motoneuron disease without polyubiquitinated inclusions have been described[43] and they seem to correlate with forms of motoneuron disease with prolonged survival, or with unusual familial cases of progressive muscular atrophy[43].

NMJs might be particularly vulnerable to defects in synaptic vesicle turnover due to their large presynaptic size and large number of synaptic vesicles compared to other synapses[44]. This idea is supported by the lack of behavioral deficits in Plekhg5-deficient mice and by the relatively preserved morphology of synapses in hippocampus, cerebral cortex and cerebellum.

Azzedine et al.[19] had previously characterized Plekhg5-deficient mice until 12 months of age, and had not detected moto-neuron loss at this stage. In our model, symptoms of motoneuron disease did not start before 12 months (Fig. 1c, e–g). Furthermore, motoneuron loss was not detectable before 12 months of age, and it progresseed until 24 months (Fig. 1d). Therefore, it is possible that motoneuron loss would have become apparent in the model described by Azzedine et al.[19] in mice older than 12 months. Furthermore, different strategies for generation of the null-allele could contribute to the severity of disease. The model described by Azzedine et al.[19] had been generated by insertion of loxP sites flanking exon 9–13. Excision of this target region results in a truncated protein lacking the RhoGEF domain but still containing the PH domain. In our model, the protein lacks both domains.

Our findings add to previous reports that neuron-specific disruption of autophagy by depletion of *Atg5* or *Atg7* results in progressive degeneration of motoneurons[8, 9] and that missense mutations in the autophagy receptors p62/SQSTM1 and opti-neurin are causative for ALS[45, 46]. Both proteins are substrates of Traf family member-associated NF-kappa-B activator (TANK)-binding kinase 1 (TBK1). Upon phosphorylation, p62/SQSTM1 and optineurin are recruited to mitochondria during PTEN-induced putative kinase protein 1 (PINK1)/Parkin-mediated mitophagy[47], a mechanism implicated in the pathogenesis of Parkinson's disease[48]. Previous reports on *TBK1* mutations in ALS[49, 50] and the recently identified interaction of the putative Rab GTPase GEF chromosome 9 open reading frame 72 (C9ORF72) with WD repeat-containing protein 41 (WDR41) and Smith-Magenis syndrome chromosomal region candidate gene 8 protein (SMCR8)[51] support the idea that impaired autophagy could make major contributions to the pathophysiology of motoneuron disease. Our study provides a first mechanism to explain how autophagy of synaptic vesicles is regulated in axon terminals of motoneurons and adds further weight to the hypothesis that dysregulated autophagy constitutes a major pathogenic mechanism in motoneuron disease.

## Methods

**Statistics**. Whiskers in box–whisker plots show minima and maxima. Boxes extend from the first to the third quartiles with cross lines at the median. Means are depicted by plus signs. In dot plots, dots represent individual experiments with cross lines at the mean ± SEM. When comparing two groups, statistics were performed using two-tailed Student's *t*-tests for unpaired samples assuming unequal variance. When comparing multiple groups, one-way ANOVA or two-way ANOVA (grouped analysis) was performed. *P* values below 0.05 were considered significant (\**P* < 0.05; \*\**P* < 0.01; \*\*\**P* < 0.001). For multiple comparisons of individual groups, the Bonferroni post-test was performed. No statistical tests were used to predetermine sample sizes, but our sample sizes are similar to those generally employed in the field. Normal distribution of data was assumed, but not formally tested. All statistical analyses were performed using GraphPad Prism 6.00.

**Generation of transgenic mice**. To generate Plekhg5-deficient mice, embryonic stem cells (ESCs) with a disruption of one *Plekhg5* allele from gene-trap clone E277F07 (cell line: E14TG2a.4 (ES cells 129P2 (formerly 129/Ola))) were injected into blastocysts of B6D2F2 (C58bl6/DBA2) mice[52] and transferred into pseudo-pregnant NMRI-foster mothers. Correct insertion site of the gene-trap cassette was verified by sequencing. Offspring were genotyped by PCR using the following primers: a 5′primer located in the intron upstream of the GT-Cassette (5′ TAAAAGCTGGCAGCCTGAAT 3′), a 3′ primer located in the GT-Cassette (5′ GCTAGCACAACCCCTCACTC 3′), and a 3′primer located in the intron downstream of the GT-Cassette (5′ ACCCCAAGGTCTGTCCTCTT 3′).

Mice were backcrossed for at least four generations onto C57Bl/6 background. Genders within genotypes were grouped, as differences between them were not statistically significant. The age of mice used for individual experiments is indicated accordingly throughout the results section.

This study was approved by the local veterinary authority (Veterinaeramt der Stadt Wuerzburg) and Committee on the Ethics of Animal Experiments, i.e., Regierung von Unterfranken, Wuerzburg (License number 55.2-2531.01-8/14).

**RNA extraction and RT-PCR**. Dissected tissues were frozen in liquid nitrogen and RNA was extracted using standard phenol–chloroform extraction. 1 μg of total RNA was treated with DNase I (Fermentas) and reverse transcribed with First Strand cDNA Synthesis Kit (Fermentas). cDNA was diluted 1:5 and 1 μl was subsequently used as template for RT-PCR reactions.

The following primers were used to validate gene-trap cassette functionality. According to www.ensemble.org, the murine Plekhg5 locus is can generate four different protein-coding transcripts. Specific 5′ primers were used for each transcript: Isoform 001 and 004: 5′ GAGCTACCCTCCCCAGACTC 3′ (ENSMUSE00000523883), Isoform 003: 5′ CTGCAGAGGAGAAGGGACTG 3′ (ENSMUSE00000594961), Isoform 002: 5′ GGTACCTAGAGGCCCGAAAC 3′ (ENSMUSE00000667027). A common 3′ primer was used for the wild-type-allele: 5′ GGTGCTGTGGAACTTGCTATC 3′ (ENSMUSE00000594961). The following 3′ primer was used to detect expression of the gene-trap-allele: 5′ GGCCTCTTCGCTATTACGC 3′.

Using 3′ primers specific for exons 6 and exon 9, we confirmed that insertion of the GT-cassette did not disturb the normal splicing pattern by altering the order of 5′exons or elimination of 5′ exons. The following 3′ primer specific to exon 6 (ENSMUSE00000594959) was used: 5′ GGCTGGCACAATTTCTGTTT 3′. The following 3′ primer specific to exon 9 (ENSMUSE00000594955) was used: 5′ GCAGAGAACAGCTGGAAGGT 3′.

**Grip strength measurements**. Force measurements were performed on a thin metal mesh connected to a force meter detecting forces within a range of 0–200 cN. Mice were suspended and allowed to hold on the grid, and, after they had grasped the grid using both paws, a force was applied to pull the mice from the grid. The force required to detach the mice from the grid was recorded, and the mean value from five attempts was taken as one data point and considered as the grip strength.

**Nissl staining and quantification of spinal motoneurons**. Mice were deeply anaesthetized and trans-cardially perfused with 4% PFA. 12.5 μm paraffin serial sections of the spinal cord were produced for Nissl staining, as described previously[53]. Cresyl violet intensively stains acidic structures within cells, and renders the nucleolus and the rough endoplasmic reticulum visible. Only motoneurons with a clearly distinguishable nucleolus, that had Nissl-stained rough endoplasmic reticulum-like structure in the cell body were counted in every tenth section of the lumbar spinal cord (L1–L8). Raw counts were corrected for double counting of split nucleoli as described[53].

**Succinate dehydrogenase enzyme activity**. Succinate dehydrogenase (SDH) stain was used to distinguish between oxidative and glycolytic fibers on 10 μm thick cryosections of tibialis anterior leg muscles. SDH staining was performed on fresh, untreated 10 μm cryosections that were incubated for 5 min in the dark at room temperature in standard SDH-substrate solution (0.2 M Na-succinat, 10 mM KCN, 10 mM nitro-blue tetrazolium, 2 mM phenazin methosulfate in phosphate buffer at pH 7.2). Reaction was stopped with 3,7% buffered formaldehyde. Stained sections were photographed under identical conditions.

**Immunohistochemical staining of neuromuscular junctions**. Staining of NMJs was carried out as previously described[54]. Briefly, mice were deeply anaesthetized and trans-cardially perfused with 4% paraformaldehyde (PFA). Subsequently, the muscles were dissected and post-fixed in 4% PFA for at least 2 hours. The tissue was washed in PBS-T (0.1% Tween-20) for 20 min at room temperature (RT) and incubated with ω-Bungarotoxin-Alexa-488 (Invitrogen) for 25 min at RT. The tissue was then incubated overnight at 4 °C with a blocking solution (2% BSA, 0.1%

Tween-20 and 10% donkey serum), followed by incubation with the primary antibodies for 3 days at 4 °C. After washing with PBS at pH 7.4 (PAA Laboratories) thrice for 15 min, the appropriate secondary antibodies were applied for 1 h at RT. The tissue was washed again as above, and embedded in Aqua Polymount (Polysciences). The following primary antibodies were used: anti-NF-H (1:5000; Millipore, AB5539), anti-Synaptophysin-1 (1:500; Synaptic Systems, 101 004). Alexa-647 and Alexa-546-conjugated secondary antibodies were from Jackson Immuno-Research Laboratories.

**Immunofluorescence staining of free-floating sections**. Mice were deeply anaesthetized and trans-cardially perfused with 4% PFA. After removal of the spinal cord, spinal cord was post-fixed in 4% PFA overnight and subsequently incubated in 30% sucrose at 4 °C overnight. Thirty-five micrometers of thick free-floating sections were cut on a Leica 9000 s sliding microtome as described previously[62] and collected in 0.1 M phosphate buffer (PB) pH 7.4. After incubation of the sections for 1 h with 4% normal goat or donkey serum and 0.3% Triton X-100 for blocking of nonspecific binding at room temperature, sections were incubated overnight at 4 °C with primary antibodies in blocking solution. After three times 10 min washing in 0.25% Triton X-100 in PB at room temperature, sections were incubated in with fluorescently labeled secondary antibodies, washed again and finally mounted with Mowiol/DABCO. The following primary antibodies were used: anti-NeuN (1:1000; Millipore, MAB377, clone A60), ChAT (1:1000; Millipore, MAB144P), anti-Ubiquitin (1:500; DAKO, Z0458), anti-Synaptophysin-1 (1:500; Synaptic Systems, 101 004). Alexa-647-, Alexa-488-, and Alexa-546-conjugated secondary antibodies were from Jackson Immuno-Research Laboratories. For visualization of F-actin Alexa Flour 532 conjugated Phalloidin (Invitrogen) was used.

**Electron microscopy**. Mice were anesthetized and trans-cardially perfused according to local institutional guidelines in 3 steps, slightly modified according to Forssmann et al.[55] with 3% paraformaldehyde, 3% glutaraldehyde, 0.5% picric acid in 0.1 M sodium phosphate buffer, pH 7.2 for 10 min. Dissected organs were fixed in the same solution for additional 1–2 h at 4 °C, post-fixed in buffered 2% osmium tetroxide (2.5 h, 4 °C), and embedded in Araldite.

To analyze the ultrastructure of NMJs, animals were trans-cardially perfused under deep anesthesia for 40 s with heparin/procaine followed by 2% formaldehyde and 4% glutaraldehyde in 0.1 M cacodylate buffer with 2 mM $MgCl_2$ and 3 mM $CaCl_2$ for 10 min. Dissected organs were further fixed for 2–4 h at 4 °C and post-fixed with 2% $OsO_4$ and 0.15% potassium hexacyanoferrate(III) to enhance visibility of the extracellular matrix.

For identification of target regions by optical microscopy, 1.5 µm-thick sections were stained with Richardson's blue (1% w/v methylene blue, 1% w/v Azur II) for 3 min at 80 °C. 60–80 nm sections (stained for 40 min in uranyl acetate and 8 min in lead citrate) were used for electron microscopy (Zeiss EM 109).

**Primary motoneuron culture**. Murine embryonic spinal motoneurons were isolated and cultured as described[56]. Briefly, after dissection of the ventrolateral part of E12.5 embryos, spinal cord tissues were incubated for 15 min in 0.05% trypsin in Hank's balanced salt solution. Cells were triturated and incubated in Neurobasal medium (Invitrogen), supplemented with 1× Glutamax (Invitrogen) on Nunclon plates (Nunc) pre-coated with antibodies against the p75 NGF receptor (MLR2, kind gift of Robert Rush, Flinders University, Adelaide, Australia) for 45 min. Plates were washed with Neurobasal medium, and the remaining motoneurons were recovered from the plate with depolarization solution (0.8% NaCl, 35 mM KCl and 2 mM $CaCl_2$) and collected in full medium (2% horse serum, 1× B27 in Neurobasal medium with 1× Glutamax). After counting, cell number was adjusted to 1000 in 100 µl, and 1000 cells were plated on four-well dishes (Greiner, Cellstar) pre-coated with poly-ornithine/laminin (Invitrogen). Cells were cultured in the presence of the neurotrophic factor BDNF. For survival assays, cells were counted 4 h after plating to find the total number of plated cells. Cells were counted again after 5 and after 7 days in vitro (DIV).

For lentiviral transduction, motoneurons were incubated with viral particles for 10 min at RT directly before plating.

**Culture of neurospheres**. The forebrain was dissected from 11.5 to 12.5-day-old mouse embryos and transferred to 100 µl HBSS. After treatment with trypsin (Gibco; 0.05%, 15 min), cell suspensions were generated by trituration. Trypsin was inactivated with egg yolk sack trypsin inhibitor (Sigma; 0.05%), and cells were plated on 75 ml flasks (Greiner) in 5 ml Neurobasal medium (Invitrogen) containing Glutamax (1:100), B27 supplement (Invitrogen), basic fibroblast growth factor (bFGF), and epidermal growth factor (EGF) (Peprotech), each at a final concentration of 10 ng/ml. Cells were passaged at least once before being used in the first experiment. For differentiation, neurospheres were dissociated using 0.05% trypsin and plated on Poly-L-ornithine-coated dishes at a density of 20,000 cells per $cm^2$. One day after plating, cells were transferred to an EGF and bFGF-depleted medium, and cultured for the indicated time intervals.

**Immunocytochemistry**. For immunocytochemistry, cells grown on glass coverslips were fixed with buffered PFA for 20 min at RT, washed three times with PBS for 15 min, and blocked for 30 min with blocking buffer containing 10% donkey serum

and 0.3% Triton X-100 in TBST (TBS-Tween). After three washes with TBST for 5 min at room temperature, cells were incubated with primary antibodies overnight at 4 °C, followed by three washes in TBST at RT for 15 min, and then incubated with the appropriate fluorophore-conjugated secondary antibodies for 1 h. Subsequently, cells were washes three times with TBST for 15 min. The coverslips were mounted on glass slides with Aqua Polymount. Cells were imaged using an Olympus Fluoview 1000i confocal microscope. The following primary antibodies were used: anti-phospho-Tau (Ser199/202) (1:500; Sigma-Aldrich, T6819), anti-Rab26 (1:500; Synaptic Systems, 269 011, clone 163E12), anti-Synaptophysin-1 (1:500; Synaptic Systems, 101 004). Alexa-647-, Alexa-488- and Alexa-546-conjugated secondary antibodies were purchased from Jackson Immuno-Research Laboratories. For visualization of F-actin Alexa Flour 532 conjugated Phalloidin (Invitrogen) was used.

**Plasmid construction**. FUW-RFP-GFP-LC3 was generated by digestion of the FUWG plasmid by Xba and EcoRI, and the resulting 10 kB fragment was purified by gel extraction. pmRFP-GFP-rLC3[57] was digested by NheI and EcoRI, and the resulting insert was purified by gel extraction and cloned into the FUW backbone by ligation. The XbaI site was destroyed during cloning.

For generation of FUW-EGFP-Rab26-WT and FUW-EGFP-Rab26-QL[15], pEGFP-Rab26-WT, and pEGFP-Rab26-WT were digested by BamHI and NheI and the resulting fragments purified by gel extraction. Subsequently, both inserts were ligated into the BamHI and XbaI sites of FUWG.

For generation of FUW-mRFP-GFP-LC3-CMV-Flag-Rab26, pCMV-Tag2a-Flag-Rab26 was digested by SalI and the resulting CMV-Flag-Rab26 fragment was purified, blunted and phosphorylated. FUW-RFP-GFP-LC3 was digested by EcoRI and blunted. Subsequently, both fragments were ligated.

To generate Plekhg5 knockdown constructs, different short hairpin (sh)-sequences were synthesized as sense and antisense oligos, and cloned into the BamHI and EcoRI restriction sites of pSIH-H1-eGFP. The following sequences were validated for their ability to knockdown Plekhg5: sh-A: 5′ TCAAGTCGGTGCTAAGGAA 3′; sh-B: 5′ ATAGCAAGATGGACGTGTA 3′; sh-C: 5′ GGACACTATTTACAACGCA 3′; sh-D: 5′ GGACGAATCTT CTCTCAGT 3′; sh-E: 5′ CGCAAGAACATGTCTGAAT 3′. Sh-D was used in the experiments, if not indicated otherwise.

The DH-PH domain of Pleghg5 was generated by amplifying the coding DNA from cDNA of human Plekhg5 (BC042606) as a template using following primers (uppercase letters indicate gene-specific nucleotides): 5′ atctggttccgcgtggatccGATGGGCATGAGAAGCTG 3′, 5′ tcacgatgcggccgctcgagTCACTGTGCACGCAGCTG 3′. The PCR product was cloned into the pGEX-4T vector using the Gibson assembly kit (New England Biolabs) according to the manufacturer's protocol.

GST-hRab26WT and GST-mRab5WT were cloned into pGEX-2T as described.

**Lentivirus production**. Lentivirus was produced by co-transfecting HEK 293T cells with the indicated expression and packaging plasmids using Lipofectamine 2000 (Invitrogen). The medium was replaced 24 h after transfection and collected 24 h later. Subsequently, the virus was concentrated by ultracentrifugation.

**Western blotting**. Equal amounts of protein were separated by SDS-PAGE, and transferred to PVDF or nitrocellulose membranes (Pall). Membranes were blocked in TBST with 5% milk powder for 1 h at RT, probed with primary antibodies overnight at 4 °C, and incubated with horseradish peroxidase-conjugated secondary antibodies for 1 h at RT. The following primary antibodies were used: anti-LC3 (1:2000; Novus Biologicals, NB100-2220), anti-Calreticulin (1:8000; Thermo Scientific, PA1-902A), anti-Calnexin (1:8000; Enzo, ADI-SPA-860), anti-HSP90 (1:4000; Enzo, ADI-SPA-830-D, clone AC88), anti-HSP70 (1:2000; Cell Signaling, 4872 T), anti-BiP (1:2000; Cell Signaling, 3177P, clone C50B12), anti-CHOP (1:1000; Cell Signaling, 2895, clone L63F7), anti-PERK (1:1000; Sigma-Aldrich, P0074), anti-phospho-Perk (Thr980) (1:1000; Cell Signaling, 3179, clone 16F8), anti-Tuj1 (1:4000; Neuromics, MO15013), anti-Flag (1:10,000; Sigma-Aldrich, F7425), anti-Actin (1:8000; Millipore, MAB1501R), anti-Gapdh (1:8000; Calbiochem, CB1001, clone 6C5), anti-IRE1α (1:1000; Cell Signaling, 3294), anti-Ubiquitin (1:1000; Enzo, BML-PW0930, clone P4D1), anti-Rab26 (1:1000; Cell Signaling 269 011, clone 163E12), anti-Synaptophysin-1 (1:1000; Synaptic Systems, 101 004), anti-Synapsin-1 (1:4000; Synaptic Systems, 106 103), anti-Syntaxin 1A (1:1000; Synaptic Systems, 110 111, clone 78.3), anti-Synaptotagmin-1 (1:1000; Synaptic Systems, 105 011,clone 41.1), anti-Synaptobrevin-2 (1:1000; Synaptic Systems, 104 211, clone 69.1), anti-Snap 25 (1:1000, Synaptic Systems, 111 011, clone 71.1) Appropriate peroxidase-conjugated secondary antibodies were from Jackson Immuno-Research Laboratories.

**Culture and transfection of NSC34 cells**. NSC34 cells were cultured in Dulbecco's modified Eagle's medium (DMEM, Gibco) containing 10% FCS, 1× Glutamax and 1× Penicillin/Streptomycin and transfected with Turbofect (Life Technologies). For pull-down experiments, 1 day prior to transfection $2 \times 10^6$ cells were plated on 6 cm dishes and transfected using 8 µg plasmid and 24 µl Turbofect. For starvation experiments $1 \times 10^6$ cells were plated on six-well plates one day before transfection. The next day, cells were transfected using

4 μg plasmid and 12 μl Turbofect. Transfections were carried out according to the manufactures instructions. All subsequent assays were performed 72 h after transfection.

**Lipid beads pull-down assay**. Lipid beads pull-down assays were performed as described[58], with minor modifications. NSC34 cells transiently expressing Plekhg5-Flag were suspended in lipid-binding buffer (20 mM Tris-HCl, 150 mM NaCl, and 1 mM EDTA, pH 7.5). The cells were extruded 10 times through a G25 syringe needle and sonicated on ice. After removal of insoluble debris by high-speed centrifugation at 13,000×g for 1 h at 4 °C, a 50 μl slurry of PI(3)P-conjugated or unconjugated beads (Echelon Bioscience) was added to the tube and rotated for 2 h at 4 °C. After washing the rotating beads five times for 20 min at 4 °C with lipid-wash buffer (10 mM HEPES, pH 7.4, 150 mM NaCl, 0.25% NP-40), bound proteins were eluted by boiling in 2× Laemmli buffer.

**Metabolic labeling, IP and thin layer chromatography**. $10^6$ cells from dissociated neurospheres were plated on PORN-coated 10 cm dishes. A day after plating, the growth medium was replaced by bFGF and EGF-depleted medium. Three days after plating, cells were transduced with FUW-Rab26-WT and cultured for additional 96 h. Thirty minutes before $^{32}$P labeling, cells were switched to phosphate-free medium. Subsequently, cells were labeled for 4 h in phosphate-free medium containing 1 mCi $^{32}$Pi in a volume of 6 ml phosphate-free medium. After labeling, cells were washed once with 5 ml ice-cold PBS and placed in 500 μl lysis buffer (50 mM Tris-Cl, pH 7.5, 150 mM NaCl, 5 mM MgCl$_2$, 1% Triton X-100, 0.5% CHAPS supplemented with protease inhibitors). Plates were kept on ice for 10 min, and then centrifuged for 3 min at maximum speed. A lysate aliquot was retained for Western blotting to test loading similarity of the samples.

Supernatants were transferred to microfuge tubes containing 30 μl GFP antibody conjugated to agarose (clone RQ2, MBL) and incubated for 2 h at 4 °C while mixing. Beads were then washed five times in ice-cold lysis buffer and nucleotides bound to the GTPase were eluted in 20 μl elution buffer (2 mM EDTA, 1 mM GTP, 0.2% SDS, 5 mM DTT) at 65 °C for 5 min. Ten microlitre eluate was spotted on 20 × 20 cm polyethylenimine (PEI) cellulose plates (Macherey-Nagel) and separated in a chromatography chamber saturated with 0.75 M KH$_2$PO$_4$ (pH 3.4). Chromatography chambers were prepared one day in advance by filling the bottom with 50 ml potassium phosphate solution. GDP and GTP were separated by placing the dry TLC sheet upright in a sealed chamber until the solvent has ascended to 70% of its length. After drying, TLC plates were exposed to an X-ray film for 3–5 days.

**Protein expression and purification**. The GST-tagged DH-PH domain (amino acids 362–748) of human Plekhg5 protein, GST-tagged human Rab26-WT and GST-tagged mouse Rab5-WT were expressed in BL21 (DE3) E. Coli. One litre of Terrific Broth supplemented with salt and 100 mg/ml of ampicillin was inoculated with 200 ml of an overnight (o.n.) culture for 3 h at 37 °C to an OD of 0.8. Subsequently, expression was induced by adding 0.2 mM IPTG and bacteria were incubated o.n. at 22 °C. Bacteria were pelleted at 4000 r.p.m. for 15 min. Pellets were washed in 1× PBS, centrifuged again as above, and resuspended in Plekhg5 (20 mM HEPES pH 7.4, 500 mM NaCl, 1 mM DTT, 5 mM EDTA and 10% gly-cerol), or Rab26/Rab5 (20 mM Tris pH 7.4, 500 mM NaCl, 5 mM MgCl$_2$, 100 μM GDP and 5 mM DTT) protein buffer. Cell suspensions were lysed with lysozym (Roth), supplemented with inhibitor cocktail (Roche), 1 mg/l DNase (Applichem), and 1% Triton X-100. Lysates were sonicated (Branson Sonifier 450) four times for 30 s. In between, lysates were and kept on ice for 30 s. Lysates were pre-cleared by centrifugation at 13,000 r.p.m. for 45 min. Supernatants were filtered through a 0.45 μm (Whatman) filter, and rotated with glutathione conjugated to sepharose (Amersham Bioscience) for 2 h at 4 °C. After collecting the flow-through, the bound fraction was washed with 1 l of protein buffer supplemented with 1 mM ATP and 0.1% Triton X-100 for the DH-PH domain of PlehG5, 1% Triton X-100 for Rab26, or no additive for Rab5. Subsequently, Rab5 and Rab26 samples were digested in protein buffer supplemented with 150 mM NaCl and 5 mg/ml thrombin o.n. at 4 °C. The Rab5 sample was loaded on a Superdex 75 16/60 column (GE healthcare). The GST-DH/PH domain was eluted by 30 mM free and reduced glutathione. The beads were incubated four times with 10 ml of elution buffer for 10 min. Fraction were examined by SDS-PAGE. The protein-containing fractions were combined and dialyzed in protein buffer containing 150 mM NaCl and 0,1% Triton X-100 o.n. Purified proteins were snap frozen and stored at −80 °C.

For Rab27B and Rab33B the beads were washed with 500 ml wash buffer containing 20 mM HEPES pH 7.4, 500 mM NaCl, 5 mM MgCl$_2$ and 50 mM HEPES pH 7.5, 200 mM NaCl, 5 mM MgCl$_2$, respectively.

The proteins were eluted in their respective wash buffer supplemented with 400 mM imidazole 4–5 times with 7 ml buffer each elution fraction. Then the proteins were dialyzed with wash buffer containing reduced salt concentration of 150 mM NaCl. Rab33B was dialyzed in wash buffer containing 30 mM HEPES. All the purified proteins were snap frozen and stored at −80 °C for downstream experiments such as GEF assay.

**Measurement of rapid kinetics**. Nucleotide binding to Rab GTPases was analyzed by stopped flow instrument (SX-20 MV, Applied Photophysics) using Mant-GppNHp (Jena Bioscience GmbH). An aliquot of 100 nM Mant-GppNHp, either in the presence or absence of 10 μm Plekhg5, was rapidly mixed with 1 μm Rab GTPase in 10 mM MgCl$_2$, 20 mM HEPES, pH 7.4, 150 mM NaCl, 1 mM DTT and 5% glycerol. Mant-GppNHp was excited at 366 nm and fluorescence emission was monitored using a passing cut-off filter (KV395, Schott).

**Electrophysiological analyses**. For the electrophysiological tests six wild-type (3 female, 3 male) and six knockout (2 female, 4 male) animals were used. Electrophysiological experiments were carried out in accordance with the internationally accepted principles in the care and use of experimental animals, approved by the local Institutional Animal Care and Research Advisory Committee and permitted by the local government (Lower Saxony, Germany; AZ 13/1070). Mice were housed under controlled conditions in the Central Animal Facility of Hannover Medical School for nearly one week before measurements. Animals were anaesthetized by isoflurane (1.5–2% with pure oxygen (1 l/min); Baxter AG, Unterschleißheim, Germany). Prior to the electrophysiological measurements Carprofen (0.01 ml/100 g; 5 mg/kg; Rimadyl, Pfizer GmbH, Karlsruhe, Germany) was given subcutaneously for analgesia, and dexpanthenol eye ointment (Bepanthen Augen-und Nasensalbe, Bayer Vital GmbH, Leverkusen, Germany) was applied to the eyes to prevent corneal dehydration. During measurements, body temperature of the animals, monitored and recorded rectally, was maintained between 33 and 37 °C by using a thermostat heating plate (direct current). Portable EMG equipment (Natus Keypoint Focus, Natus Europe GmbH, Planegg, Germany) along with the Keypoint.net (version 2.32) computer program was used. For nerve conduction studies, the sciatic nerve of ~24-month-old animals was stimulated percutaneously by single pulses of 0.1 ms and 1 Hz delivered through a pair of needle electrodes (Spes Medica disposable monopolar needle electrode, 13 mm × 33 G, GVB geliMED KG, Bad Segeberg, Germany) placed proximally at the sciatic notch and distally at the popliteal fossa. The distance between both stimulation electrodes was measured. Reference electrodes were inserted subcutaneously a few millimeters apart from the stimulation electrodes. The maximal compound muscle action potential (CMAP) was recorded using needle electrodes (same as for stimulation) at the gastrocnemius, anterior tibial or plantaris muscles. The reference electrode for all three muscles was placed in the fourth toe. The ground electrode was inserted subcutaneously rostral to the stimulation electrodes. Amplitude (baseline-peak, mean proximal/ distal) and latency (time from stimulus to the onset of first negative deflection, mean proximal/ distal) were measured. Nerve conduction velocity (NCV) was calculated via the latency differences of the proximal and distal stimulation and the stimulation distance (mean NCV per muscle gastrocnemius/ tibialis). Motor unit number estimation (MUNE) was performed subsequently to the nerve conduction study, based on the description in Arnold et al.[59]. At least ten single motor unit potentials (SMUPs; baseline-peak amplitude) were obtained by incremental stimulation technique (1 Hz, 0.1 ms, 0.02 mA increase). Only increases over 25 μV in the amplitude were recorded trying to minimize the influence of alternation. The incremental values of the SMUPs were averaged in order to determine the size of the mean SMUP. In the end, the estimated number of motor units within the tibialis and gastrocnemius muscle was calculated by dividing the respective mean SMUP into the maximum CMAP amplitude.

**Mechanical sensitivity**. To determine mechanical paw withdrawal thresholds, we used the von Frey test based on the up-and-down-method of Chaplan et al.[60]. Experimental mice were placed on a wire mesh in individual acrylic-glas cages and the plantar surface of the hind paws was alternately touched with a calibrated von Frey filament applying mild pressure leading to a slight bending of the filament; the experiment was started at 0.69 g. Upon hind paw withdrawal the next thinner von Frey filament was used. If the hind paw was not withdrawn, the next thicker von Frey filament was used. Each hind paw was assessed three consecutive times and the 50% withdrawal threshold (i.e. force of the von Frey hair to which an animal reacts in 50% of the administrations) was calculated.

**Heat sensitivity**. To determine paw withdrawal latencies upon heat stimulation we applied a standard algometer (Ugo Basile, Gemonio, Italy) based on the method of Hargreaves et al.[61]. Experimental mice were placed on a glass surface in individual plexiglas cages and a radiant heat source (25 IR) was placed under one hind paw each. Time until paw withdrawal was recorded automatically. Heat application was limited to 16 s to avoid tissue damage. Each hind paw was tested three times consecutively.

**Behavior**. Plekhg5-deficient (n = 10) and control mice (n = 11), at age of 12–14 months with no symptoms of motoneuron disease were used for behavioral analysis. Genders within genotypes were grouped, as differences between them were not statistically significant. Mice were housed individually under a 12 h light/dark cycle (6:00 a.m.–6:00 p.m.) with ad libitum access to food and water. The cages (Tecniplast, 1264 C Eurostandard Typ II, 267 × 207 × 140 mm) were kept in a Scantainer (Scanbur Ltd. Denmark) assuring stable conditions through a constant airflow and maintaining a temperature of about 21 °C and air humidity of about

55%. Experimental procedures took place during the light phase of the cycle between 8:00 a.m. and 5:00 p.m. Mice were transported in their home cage to a separate room and allowed to calm down for half an hour. Then mice were taken individually in their home cages to the experimental room for experiments. The experimenter was unaware of the genotype to keep experiments unbiased.

For the Open Field test, mice were placed in the middle of an open field arena (48 × 48 cm, height 50 cm, evenly illuminated to ~ 40 lux) and filmed for 10 min with a webcam-based system (Logitech). Movements of the mice were tracked with VideoMot2 (TSE, Germany) and the following parameters were analyzed: time spent in center, number of entries into center. The area was cleaned thoroughly with water after each trial.

The Object Recognition experiments took place in the open field arena. For the object recognition task two kinds of objects were used: a cell culture flask and a tower of about the same size built of differently colored Lego bricks. The objects were placed in diagonally opposing corners of the arena with 12 cm distance to from both sides of the wall. On day 1 half of the animals were presented with two cell culture flasks, the other half with two Lego towers. Mice were placed in the middle of the arena and allowed to explore the arena and the objects for 10 min. On day 2 one of the objects was replaced by a novel object (either cell culture flask or Lego tower) at either position 1 or 2 in a pseudo-randomized manner to avoid position bias. Each mouse was therefore confronted with a familiar and a novel object and was allowed to explore freely for 10 min. Mice were automatically tracked (see above). We scored exploratory behavior toward objects manually whenever the mouse sniffed the object or touched the object while looking at it. After each trial, the arena and the objects were cleaned with water.

Fear conditioning was performed in a fear conditioning apparatus (mouse multi-conditioning system, 256060 series, TSE, Germany). Fear conditioning took place in context A, which consisted of a square arena with two blackened sides placed on a metal grid. Context B, a round arena, which was placed on a black plastic ground, served as new context. To support context discrimination, mild olfactory cues were introduced into the arenas by wiping them with tissue paper moistened with 70% ethanol for context A and 1% acetic acid for context B. The conditioning consisted of two phases. On day 1 (fear acquisition), mice were exposed to context A for 180 s before a tone (CS, 80 dB, 5000 Hz) was presented for 10 s and co-terminated with an electric foot shock of 0.7 mA for 1 s (US). The CS and the US were presented three times with an inter-stimulus-interval (ISI) of 20 s. After the last presentation, mice stayed in context A for 150 s before being returned to the home cage. On day 2, mice were placed in context B to test for context discrimination. For fear retrieval, after 60 s of free exploration the CS was presented without the US for 12 times with an ISI of 20 s. After another 60 s mice were returned to their home cage. Mice were video tracked and movement was recorded by a light beam system. Freezing behavior, defined as immobility except for respiration movements, was analyzed with a software-assisted system (TSE MCS FCS) and a freezing threshold of 2 s. After each trial, the context was cleaned with 70% ethanol or 1% acetic acid, respectively.

To account for statistical differences between genotypes and conditions depending on the experiment, two-way ANOVAs with post hoc Sidak's multiple comparison test or unpaired $t$-tests were performed. Analyses were performed with GraphPad Prism 6 and are represented as $\pm$ SEM. Results were considered statistically significant at $P < 0.05$.

**Data availability**. The authors declare that all data supporting the findings of this study are available within the article and its Supplementary Information files or from the corresponding author on reasonable request.

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

## Acknowledgements

We are grateful to Sebastian van der Linde for his support with SIM microscopy. We thank Regine Sendtner for grip strength and rotarod measurements and Hildegard Troll for lentivirus production. We are grateful to Katrin Walter and Nicole Rachor for excellent technical assistance. C.S. and C.K. were supported by grants of the Fritz Thyssen Stiftung für Wissenschaftsförderung (Az. 10.14.1.189). P.L. and M.S. were supported by grants from the German Government: BMBF "DysTract" and EnergI, "Neuronal basis of healthy aging", the Deutsche Forschungsgemeinschaft Grant DFG, SE 697/5-1, the Bavarian Research Network "Human iPS Cells"(ForIPS). D2-F2412.26, the German Society for patients with neuromuscular disorders (DGM) "Analysis of cellular mechanisms for motoneuron degeneration" and the Hermann und Lilli Schilling Stiftung im Sifterverband der Deutschen Industrie.

## Author contributions

M.S., C.K., B.K., and P.L. conceived the study. P.L. performed IHC, western blot analysis, metabolic labeling and pull-down experiments (Fig. 1a, c, m; Fig. 3i, j; Fig. 5e–j; Fig. 7a, b; Fig. 8; Fig. 9; Supplementary Figs. 1, 4, 5, 6). P.L. and B.B. performed plasmid design and construction. P.L. and S.J. performed and analyzed histology (Fig. 1d, h). P.L. and B.D. performed and analyzed in vitro experiments (Fig. 5a–d; Fig. 6). P.L., B.D. and C.S. performed and analyzed NMJ stainings (Fig. 2; Supplementary Fig. 3). P.H. performed and analyzed electron microscopy with assistance of C.S and P.L. (Fig. 3a–h, Fig. 4). E.-M.F. and A.F. generated transgenic mice. N.H.-T. and S.P. performed and analyzed electrophysiological analyses (Fig. 1i–m, o). C.R.v.C. and R.B. performed and analyzed behavioral tests (Supplementary Fig. 2). F.K. and N.Ü. performed and analyzed sensory tests (Fig. 1p, q). I.M. provided critical reagents. A.H. provided critical reagents and helpful comments on the manuscript. M.D. provided critical reagents and performed IHC (Supplementary Fig. 6b). R.J. and B.B. conceived GEF assays. B.B. purified recombinant proteins. A.P.-L. and B.B. performed and analyzed GEF assays (Fig. 7d–h). P.L. assembled data and drafted the manuscript. P.L. and M.S. analyzed data and wrote the manuscript. All authors approved the final draft of the manuscript.

## Additional information

**Competing interests:** The authors declare no competing financial interests.

