## [Peer Review File · Nature Communications]

Reviewers' comments:

Reviewer #1 (Remarks to the Author):

In this manuscript, Lüningschrör et al describe the pathophysiological implications of autophagy regulation by Plekhg5 in motoneurons. Indeed, the autophagy pathway has been demonstrated to be essential for neuronal homeostasis and involved in many neurodegenerative diseases. Thus, the comprehension of molecular mechanisms of autophagy in neurons has major biomedical significance. Given the involvement of homozygous mutations of Plekhg5 in several forms of lower motoneuron disease, the study has relevance to understanding motoneuron vulnerability under pathological conditions. Using Plekhg5 knockout mice, the authors were able to link the altered autophagy of synaptic vesicles in the axonal terminals of motoneurons to compromised neuromuscular junction and decreased motor performance. At the molecular level, the function of Plekhg5 as guanine exchange factor for the GTPase Rab26, which was previously associated to autophagy of synaptic vesicles, was convincingly demonstrated. Thus, the novel pathogenic mechanism inferred for motoneuron disease involves the altered turnover of synaptic vesicles due to deficient Plekhg5-mediated autophagy resulting in aberrant axonal terminals and neuromuscular junction. Even though the study offers valuable insight regarding the role of Plekhg5 and autophagy to motoneuron function and integrity, the authors should also investigate the impact of ablating Plekhg5 in a well-established model of motoneuron disease to determine the relevance of the proposed mechanism. Finally, the paper should be considered for publication in Nature Communication after major revision addressing the following points.

Major points:

1. A previous study by Azzedine et al (Hum. Mol. Genet. 2013) failed to detect signs of lower motor neuron disease in Plekhg5 complete knockout mice as evaluated by histological analysis of motoneurons in spinal cord and neuromuscular junction. Only mild effects in motor nerve conduction velocity and Rotarod performance due to loss-of-function of Plekhg5 were reported in these mice. This observation is in sharp contrast to the results obtained in the Plekhg5 knockout mouse generated in the current study, where the authors report significant loss of motoneurons and motor deficits at 12 months, in addition to seemingly obvious effects on neuromuscular junction morphology and muscular fiber types. How do the authors reconcile this apparent discrepancy?
2. The authors report reduced lifespan of the Plekhg5 knockout mice. Please present the body weight curve of these mice. What is the cause of premature death?
3. The phenotypic and histological characterization of Plekhg5 knockout mice is focused on the motor system. This characterization should be complemented with electromyography recordings to assess muscle denervation. Also, the study lacks important behavioral tests addressing the impact of Plekhg5 deficiency on sensory and cognitive function of mice and histological analysis of major brain areas that must be included in the revised version of the manuscript.
4. In Fig. 1d, representative micrographs show loss of motoneurons in lumbar spinal cord of Plekhg5 knockout mice. What type of motoneurons are lost during aging? According to soma size, it seems that alpha-motoneurons are degenerating. If so, wouldn't this be contradictory to the fact that fast glycolytic fibers innervated by large alpha-motoneurons increase in proportion? Please discuss.
5. In Supplementary Figure 2 and 3, please provide quantification of altered neuromuscular junction morphology and axonal swelling.
6. Autophagy deficiency due to Plekhg5 loss-of-function could cause global impairment of proteostasis

in motoneurons underlying the phenotypic and histological alterations observed in the knockout mice. Thus, histological experiments for detection of ubiquitinated protein inclusions and biochemical analysis of heat shock protein family and unfolded protein response activation have to be performed to determine possible proteostasis disturbance.

7. The relevance of decreased autophagy of synaptic vesicles to motoneuron degeneration should be investigated in a classical mouse model of motoneuron disease. The mutant SOD1 mouse model of ALS has well-defined spatiotemporal course of disease progression and could be useful dissecting out the contribution of altered synaptic vesicle turnover to motoneuron vulnerability. The generation and phenotypic/histological characterization of mutant SOD1 mice/Plekhg5 knockout should take approximately 9 months and would greatly increase the impact of the manuscript.

Minor points:

1. In page 6, line 137, 'To assess whether dysregulated autophagy caused the accumulation at nerve terminals we examined the autophagic flux in Plekhg5 deficient motoneurons (Fig. 3).' Accumulation of what exactly are the authors referring to?

Reviewer #2 (Remarks to the Author):

Here, Luningschror et al, identifies the involvement of PLEKHG5 in regulating autophagy in motoneurons via its role as a GEF for Rab26, a small GTPase previously implicated in recruiting synaptic vesicles for autophagic degradation. The authors show that loss of PLEKHG5 results in the loss of motoneurons. In addition, membranous organelles appear in abnormally swollen neuromuscular junctions and is accompanied by an accumulation of synaptic vesicle proteins. Accordingly, motoneurons lacking PLEKH5 show diminished ability to form autophagosomes which would account for the unusual buildup of synaptic vesicle proteins. Overall the manuscript provides new and important insights towards our understanding of the processes governing turnover of synaptic vesicle proteins and is of importance for both targeted and general readerships.

Specific comments:

Authors mentioned that ultrastructurally membranes either showed open ends or failed to seal (Fig 2). Were these observations confirmed using serial section analyses?

Acute overexpression of wildtype and constitutively active EGFP-Rab26 was previously shown to induce massive accumulation of vesicles containing synaptic vesicle proteins. Did the authors observe such accumulations in wildtype motoneurons? Do these accumulations diminish or perhaps even disappear in the PLEKHG5^{-/-} background with the overexpression?

Experiments in Fig 4 elegantly detailed the role of PLEKHG5 as a Rab26 GEF. However, the subsequent experiments do not demonstrate the involvement of autophagy in this regard since measurements of axon growth were used as a readout rather than parameters to monitor autophagy as were used in Fig 3. Furthermore, it is not immediately clear how restoration of a pathway involved in synaptic vesicle turnover results in recovery of neurite extension. Do synaptic terminals degenerate in these cultures as were observed in the animals that then possibly lead to neurite retraction/degeneration? Or could Rab26 itself be involved in trafficking vesicles for this extension?

Minor comment:

In Fig 2h, do the multiple lanes represent lysates from sciatic nerves obtained from different animals? If so, this should be mentioned in the figure legends.

Reviewer #3 (Remarks to the Author):

The manuscript by Luningschror et al. describes the development and characterization of a Plekhg5 knockout mouse. The authors demonstrate that the homozygous knockout mouse displays slowly progressive motor neuron degeneration, characterized by decreased survival, weakening grip strength, loss of motor neurons, alterations in muscle histology, and alterations in NMJ morphology. EM analysis indicates alterations in organelle morphology at synapses, and western blots demonstrate the accumulation of synaptic proteins. As these defects suggest a possible deficit in autophagy, the authors express a tandem GFP-RFP-LC3 probe and find deficits consistent with an impairment of autophagy. The authors hypothesize that Plekhg5 may be a GEF for Rab26, and demonstrate that a fragment of Plekhg5 acts as a GEF for Rab26 but not Rab5. They then go on to show that a constitutively active form of Rab26 rescues the axon growth defect seen in Plekhg5^{-/-} neurons.

The strength of this work is the generation and careful characterization of a mouse model for Plekhg5 defects, and the demonstration of a defect in autophagy of synaptic components. This work makes a novel contribution, and should be published pending some important revisions:

Major:

1. The use of the constitutively active RAB26 construct to rescue the axon outgrowth deficit is a nice start, but the overall story would be much stronger if the authors showed that this construct also rescued the autophagy deficit observed in the Plekhg5^{-/-} neurons.
2. The colocalization of Rab26 and synaptophysin in Figs. 4 and 5S is not striking. These figures should be either strengthened, or de-emphasized – maybe the interaction is indirect or more likely transient.
3. Is it sufficient to show specificity by comparing the GEF activity of Plekhg5 between only two Rabs? Most studies perform multiple comparisons. At the least, Rab7 and Rab11 should also be assayed.
4. The blots shown in Fig. 3d are perplexing, because most labs see a robust LC3-I/LC3-II doublet, rather than a single major band for LC3-II. Can the authors include a parallel positive control, brain extract for example, showing the usual doublet? And can they comment on this unusual observation?
5. The description of the interaction of Plekhg5 with PI(3) seems minimal. Further, the logical progression leading to the conclusion described in lines 179-180 seems to be missing some key steps.
6. The presentation of the work is too succinct, lacks important information, and overstates a number of points. For example, the abstract includes broad statements that are not yet fully supported, rather than discussing the specific observations made in the paper, and their interesting and important implications. The introduction does not provide a useful summary of what is known about the links between Plekhg5 and disease. The body of the manuscript lacks a schematic of Plekhg5 that clarifies which domains were used in the biochemical experiments presented. The summary section very broadly discusses links between autophagy defects and disease, more suitable to a broad-based review, rather than discussing the observations made in this work and their more specific implications. And as noted below, a number of figures in the supplement would be more helpful if they appeared within the main text.

Minor:

1. Both the grip strength data from Fig. S2a and the staining data from Fig. S5 should be incorporated in the main text. If necessary to save space, Fig. 1d could be relocated to the supplement.
2. Is there any way to provide a quantitative analysis of the muscle histology shown in Fig. 1e?
3. Initial fusion between autophagosomes and LAMP1-positive late endosomes is generally thought to

occur distally in the axon, while full maturation to an autolysosome may only occur more proximally (lines 151-155).

4. Nutrient deprivation differentially affects neurons and cell lines. Therefore, the Plekhg5 degradation experiments performed in NSC34 cells should be repeated in primary neurons.

In summary, this is interesting, novel work that will be of high interest to those in the field. However, a number of key revisions are required prior to publication.

Reviewer #1 (Remarks to the Author):

In this manuscript, Lüningschrör et al describe the pathophysiological implications of autophagy regulation by Plekhg5 in motoneurons. Indeed, the autophagy pathway has been demonstrated to be essential for neuronal homeostasis and involved in many neurodegenerative diseases. Thus, the comprehension of molecular mechanisms of autophagy in neurons has major biomedical significance. Given the involvement of homozygous mutations of Plekhg5 in several forms of lower motoneuron disease, the study has relevance to understanding motoneuron vulnerability under pathological conditions. Using Plekhg5 knockout mice, the authors were able to link the altered autophagy of synaptic vesicles in the axonal terminals of motoneurons to compromised neuromuscular junction and decreased motor performance. At the molecular level, the function of Plekhg5 as guanine exchange factor for the GTPase Rab26, which was previously associated to autophagy of synaptic vesicles, was convincingly demonstrated. Thus, the novel pathogenic mechanism inferred for motoneuron disease involves the altered turnover of synaptic vesicles due to deficient Plekhg5-mediated autophagy resulting in aberrant axonal terminals and neuromuscular junction. Even though the study offers valuable insight regarding the role of Plekhg5 and autophagy to motoneuron function and integrity, the authors should also investigate the impact of ablating Plekhg5 in a well-established model of motoneuron disease to determine the relevance of the proposed mechanism. Finally, the paper should be considered for publication in Nature Communication after major revision addressing the following points.

Major points:

1. A previous study by Azzedine et al (Hum. Mol. Genet. 2013) failed to detect signs of lower motor neuron disease in Plekhg5 complete knockout mice as evaluated by histological analysis of motoneurons in spinal cord and neuromuscular junction. Only mild effects in motor nerve conduction velocity and Rotarod performance due to loss-of-function of Plekhg5 were reported in these mice. This observation is in sharp contrast to the results obtained in the Plekhg5 knockout mouse generated in the current study, where the authors report significant loss of motoneurons and motor deficits at 12 months, in addition to seemingly obvious effects on neuromuscular junction morphology and muscular fiber types. How do the authors reconcile this apparent discrepancy?

Azzedine et al (Hum. Mol. Genet. 2013) have characterized Plekhg5 knockout mice until 12 months of age and did not detect motoneuron loss. In this previous study, later stages have not been analyzed. In our model, symptoms of motoneuron disease do not start before 12 months, as shown in Fig. 1c, e-g. Furthermore, motoneuron loss was not detectable before 12 months of age, and it progresses until 24 months, as shown in Fig. 1d. Therefore, it is possible that motoneuron loss also becomes apparent in the model described by Azzedine et al when later disease stages are investigated. Furthermore, different strategies for generation of the null-allele could contribute to the severity of disease. The model described by Azzedine et al was generated by insertion of loxP sites flanking exon 9 – 13. Excision of this target region results in a truncated protein lacking the RhoGEF domain but still containing the PH-domain. In our model, the resulting protein lacks both domains. We have now addressed this issue in the discussion of the revised manuscript on page 15, lines 364-374, discussing these two points.

2. The authors report reduced lifespan of the Plekhg5 knockout mice. Please present the body weight curve of these mice. What is the cause of premature death?

We have now included body weight measurements in the new Fig. 1c. These data show a reduction of body weight in 18-24 months old mice. We also extended our analysis of motor deficits by performing grip strength and rotarod analyses. These new data in Fig. 1 e-g indicate that the disease progresses from hindlimbs to forelimbs, similar as in well characterized models of motoneuron disease such as pmn mutant mice or SOD1 G93A mice ^{1,2}.

Figure 1:

Plekhg5 deficient mice develop a motoneuron disease with late onset. (c) Body weight measurements of *Plekhg5* deficient- and control mice of different ages. (2-4 months, *Plekhg5*^{+/+}: n = 12, *Plekhg5*^{-/-}: n = 9; 9-15 months, *Plekhg5*^{+/+}: n = 19, *Plekhg5*^{-/-}: n = 42; 18-25 months, *Plekhg5*^{+/+}: n = 18, *Plekhg5*^{-/-}: n = 36; two-way ANOVA; Bonferroni post-test) (e, f) Grip strength measurements of fore- (e) and hindlimbs (f). (g) Rotarod performance. (9-15 months, *Plekhg5*^{+/+}: n = 9, *Plekhg5*^{-/-}: n = 10; 18-24 months, *Plekhg5*^{+/+}: n = 11, *Plekhg5*^{-/-}: n = 8; two-way ANOVA; Bonferroni post-test)

Figure 2:

(a, b) NMJs at the gastrocnemius (a) and intercostal (b) muscles stained for BTX and synaptophysin. (c, d) Quantification of presynaptic area by evaluation of synaptophysin staining and measurement of the widest expansion of presynaptic sites (three animals per genotype with at least 15 NMJs analyzed per animal; n = 3; mean \pm SEM; two-way ANOVA; Bonferroni post-test). (e) Semi-thin section of a swollen

axon terminal at an intercostal NMJ in *Plekhg5* deficient mice. c, capillary; mf, myofiber; art, artery; arrow points to erythrocyte; arrowheads point to myelinated axons; asterisk label degenerating presynapse.

To investigate whether respiratory muscles are affected in *Plekhg5* deficient mice we analyzed intercostal muscles. Neuromuscular endplates in intercostal muscles from *Plekhg5* deficient mice also show a degenerative phenotype similar as within the gastrocnemius muscle (Fig 2 b, d). This indicates that muscles necessary for respiration become denervated which could make a major contribution to premature death as observed in a well-characterized ALS-models such as SOD1 G93A³.

*3. The phenotypic and histological characterization of *Plekhg5* knockout mice is focused on the motor system. This characterization should be complemented with electromyography recordings to assess muscle denervation. Also, the study lacks important behavioral tests addressing the impact of *Plekhg5* deficiency on sensory and cognitive function of mice and histological analysis of major brain areas that must be included in the revised version of the manuscript.*

We have now complemented the phenotypic characterization of *Plekhg5*-deficient mice with histological analysis of major brain areas. The overall morphology and layering of the major brain areas hippocampus, cerebellum and cortex appeared normal (Supplementary Fig. 4). In contrast to the abnormally swollen appearance of motoneuron terminals with Synaptophysin-accumulations, synapses within the brain were unaffected in *Plekhg5* deficient mice (Supplementary Fig. 4). These observations confirm our data on normal cognitive function in *Plekhg5* deficient mice. However, we also observed axonal swellings in projection neurons within the spinal cord (Fig. 4h-i), indicating that other neurons with long projections and potential role in the motor system are affected. In summary, we conclude that *Plekhg5* predominantly functions in maintenance of the motor system, which would be consistent with the clinical observations in patients with mutations in the *PLEGHG5* gene.

As suggested by the reviewer, we also conducted behavioral tests for assessment of cognitive function in 12-14 old mice, before motor disease symptoms occur. *Plekhg5* deficient mice do not show abnormal behavior in the open field test (a, b) indicating that anxiety-related behavior is not altered. They also do not show differences in object recognition (c, d) and fear conditioning (e, f) indicating that learning and memory are unaffected in *Plekhg5* deficient mice.

Supplementary Figure 2

Plekhg5 deficiency does not impair cognitive function. (a, b) Control- and *Plekhg5* deficient mice were placed in the center of the open field arena and their movement was monitored for 10 minutes. (a) Representative trajectories of control- and *Plekhg5* deficient mice show the behavior of both groups in the open field test. (b) *Plekhg5* deficient mice spent the same amount of time in the center as compared to control animals. Both groups also showed the same number of entries into the center. (c, d) To assess the memory skills of *Plekhg5* deficient mice the object recognition test was performed. (c) The first day two identical objects were presented and placed in two diagonally opposing corners of the box. Mice were placed in the middle of the box and monitored and tracked for 10 minutes. The next day one of the objects was replaced with a novel object. Whether the novel object is presented at position 1 or 2 was randomized to avoid position bias. Again, mice were placed in the middle of the box and monitored and tracked for 10 minutes. (d) At day 1, control- and *Plekhg5* deficient mice spent half of the time at both objects. The second day both groups spent significantly more time at the novel object suggesting that both groups are able to recall the familiar object and recognize the new object. (e, f) We investigated fear acquisition and context discrimination of *Plekhg5* deficient mice in a paradigm of fear conditioning. (e) At day 1, mice were acquired to associate a tone with a foot shock in context A. After a 180 s habituation phase, mice received a 10 s tone of which the last second is accompanied by an electric foot shock delivered via the metal grid. The stimuli are presented 3 times with an inter stimulus interval (ISI) of 20 s. At day 2, mice have to recall the acquisition in context B. After a 60 s habituation phase, mice receive a 10s tone that is

not accompanied by an electric foot shock. The sound is presented 12 times with an ISI of 20s. (f) Quantification of freezing time. During acquisition (context A) both groups show a significant increase in freezing time after receiving an electric foot shock. At day 2 both, control – and *Plekhg5* deficient mice are able to recall the acquisition in context B. Both groups show a significantly increase in freezing time after receiving the tone without electric foot shock.

To address the impact of *Plekhg5* deficiency on sensory function we performed the von Frey and hot plate tests. Whereas *Plekhg5* deficient mice did not show major changes in mechanical withdrawal latency, they displayed hypersensitivity to heat, indicating that sensory function is not generally reduced in these mice. We cannot explain the increased sensitivity to heat at this point. This could be caused by altered reflex mechanisms on the spinal level, or could reflect alteration in heat tolerance, which has been described in patients with ALS ^{4,5}.

Figure 1

***Plekhg5* deficient mice develop a motoneuron disease with late onset.** (p, q) Box- and whisker-plots show mechanical withdrawal thresholds (p) and heat withdrawal latencies (sec) (q) of naive male *Plekhg5*^{-/-} and control littermates. Mechanical withdrawal thresholds did not differ between genotypes (p). *Plekhg5*^{-/-} mice showed heat hypersensitivity compared to control littermates. (*Plekhg5*^{+/+}: n = 5, *Plekhg5*^{-/-}: n = 5; Mann-Whitney U test).

We have now complemented the phenotypic characterization of *Plekhg5*-deficient mice with histological analysis of major brain areas. The overall morphology and layering of the major brain areas hippocampus, cerebellum and cortex appeared normal (Supplementary Fig. 4). In contrast to the abnormally swollen appearance of motoneuron terminals with Synaptophysin-accumulations, synapses within the brain were unaffected in *Plekhg5* deficient mice (Supplementary Fig. 4). These observations confirm our data on normal cognitive function in *Plekhg5* deficient. In summary, we conclude that *Plekhg5* specifically functions in maintenance of neuromuscular junctions.

Supplementary Figure 4

Histological analysis of major brain areas. Brain sections of control and Plekhhg5 deficient mice were stained for Synaptophysin, Tuj1 and DAPI. The morphology and layering of major brain areas appeared normal in Plekhhg5 deficient mice. Furthermore, the synapse-morphology was unaltered compared

4. In Fig. 1d, representative micrographs show loss of motoneurons in lumbar spinal cord of Plekhhg5 knockout mice. What type of motoneurons are lost during aging? According to soma size, it seems that alpha-motoneurons are degenerating. If so, wouldn't this be contradictory to the fact that fast glycolytic fibers innervated by large alpha-motoneurons increase in proportion? Please discuss.

Based on our observation in gastrocnemius and intercostal muscles showing that axon terminals at neuromuscular endplates degenerate we conclude that the alpha motoneurons are degenerating. This is also reflected by reduced muscle strength and fiber grouping in the gastrocnemius muscle. The tibialis anterior muscle was less affected, which we consider as a possible reason for the relatively higher density of fast glycolytic fibers. We have replaced the pictures showing SDH staining from tibialis anterior muscle by similar staining of gastrocnemius muscle because this muscle is more severely affected, as shown in the new Fig. 1 I – m (and below). For comparison, MUNE analyses in the tibialis anterior muscle of the same animals as shown above show less denervation in the tibialis muscle in comparison to the gastrocnemius muscle (see below).

5. In Supplementary Figure 2 and 3, please provide quantification of altered neuromuscular junction morphology and axonal swelling.

We have now included the quantification of altered neuromuscular junction morphology in the new Fig. 1 a-d, by presenting data on the size of axon terminals, the number of NMJs showing abnormal Synaptophysin accumulation (Fig. 2 c, d) in the gastrocnemius and the intercostal muscles from 3 animals of each group. The number of axonal swellings has been quantified from semi-thin sections (10 sections per animal) of lumbar spinal cord from three wildtype and three *Plekhg5* deficient animals, and data are shown in the new Fig. 4j.

Figure 4:

Loss of *Plekhg5* impairs axonal integrity. (f - i) Longitudinal- (f, g) and cross- (h, i) sections of lumbar spinal cord indicate axonal swellings within the white matter of *Plekhg5*^{-/-} mice. (g, i) Fine structure of axonal swellings detected in *Plekhg5* deficient mice. (j) Quantification of axonal swellings in spinal cord semi-thin cross-sections. Three animals per genotype were analyzed. Each data point represents the mean of 10 sections from one animal with a distance of 100 µm between individual sections.

6. Autophagy deficiency due to *Plekhg5* loss-of-function could cause global impairment of proteostasis in motoneurons underlying the phenotypic and histological alterations observed in the knockout mice. Thus, histological experiments for detection of ubiquitinated protein inclusions and biochemical analysis of heat shock protein family and unfolded protein response activation have to be performed to determine possible proteostasis disturbance.

In order to address this point, we have stained spinal cord cross-sections and neuromuscular endplates for synaptophysin and ubiquitin to analyze the presence of ubiquitinated protein inclusions in 18 month- old symptomatic mice. Neither motoneuron soma nor axon terminals showed ubiquitin positive inclusions even in regions with accumulation of synaptophysin. As a positive control, we used SOD1 G93A mice, which showed polyubiquitinated structures in spinal cord sections. Ubiquitin staining was found both in synaptophysin- positive and -negative structures. These results indicate that the disease mechanism in *Plekhg5* is more specific and does not include general alterations in proteostasis. This data set is now included as Suppl. Fig. 5. We also probed lumbar spinal cord lysates for ubiquitin (Suppl. Fig. 5 c). These biochemical analyses also confirm that *Plekhg5* deficiency does not alter the levels of protein ubiquitination in *Plekhg5* deficient mice.

Supplementary Figure 5:

Global protein homeostasis is not impaired by Plekhhg5 deficiency. (a) Ubiquitin staining of spinal cord cross-sections from control-, *Plekhhg5* deficient- and SOD1 G93A mice. In *Plekhhg5* deficient mice we did not detect ubiquitinated protein inclusions, in contrast to SOD1 G93A mice that showed Ubiquitin accumulations in the spinal cord. Arrowheads point to Synaptophysin accumulations. Arrows point to

accumulations positive for Synaptophysin and Ubiquitin. **(b)** Ubiquitin staining of NMJs. Ubiquitinated protein inclusions were not detectable at NMJs of *Plekhg5* deficient mice. **(c)** Western blot analysis of spinal cord extracts probed for Ubiquitin. *Plekhg5* deficient mice did not any enrichment for polyubiquitinated proteins. **(d)** Western blot quantification. (Four animals per genotype were analyzed; Mean \pm SEM; unpaired t-test; two-tailed)

We also investigated the unfolded proteins response and ER stress biochemically using spinal cord extracts from age matched WT and 18-month-old *Plekhg5*^{-/-} mice. The levels of Chop and IRE1-alpha were markedly elevated in *Plekhg5*^{-/-} samples. Thus, UPR seems to be activated in *Plekhg5* deficient mice. We also tested the expression levels of heat shock protein family members HSP70 and HSP90 by Western blot analyses and found significantly decreased expression of both proteins. This correlates with our finding that ubiquitination is not increased in spinal cord and neuromuscular endplates, indicating that the defect in *Plekhg5*^{-/-} mice is restricted to synaptic vesicles and does not involve misfolding or aggregation of cytosolic proteins. Thus, *Plekhg5*^{-/-} mice represent a model for motoneuron disease without polyubiquitinated protein inclusion in which defective autophagy of synaptic vesicles stands in the center of the pathophysiology. Such cases of motoneuron disease without polyubiquitinated inclusions have been described⁶ and they seem to correlate with forms of motoneuron disease with prolonged survival, or unusual familial cases of progressive muscular atrophy⁶.

In summary, these data indicate that the pathology in *Plekhg5*^{-/-} does not include a general defect in UPR regulation, in particular not by interfering with degradation of ubiquitinated proteins.

Figure 9:

Plekhg5-depletion in SOD1 G93A motoneurons results in elevated ER-stress. **(a)** Western blot analysis of spinal cord extracts. Expression levels of HSP70, HSP90, Calreticulin and Calnexin were examined. **(b)** Quantification of Western blot shown in (a). **(c)** Western blot analysis of spinal cord extracts. Expression levels of IRE1α and Chop1 were examined. **(d)** Quantification of Western blot shown in (c) (each data point represents expression levels of one animal; Mean \pm SEM; unpaired t-test; two-tailed).

7. *The relevance of decreased autophagy of synaptic vesicles to motoneuron degeneration should be investigated in a classical mouse model of motoneuron disease. The mutant SOD1 mouse model of ALS has well-defined spatiotemporal course of disease progression and could be useful dissecting out the contribution of altered synaptic vesicle turnover to motoneuron vulnerability. The generation and phenotypic/histological characterization of mutant SOD1 mice/Plekhg5 knockout should take approximately 9 months and would greatly increase the impact of the manuscript.*

Breeding of sufficient numbers of symptomatic mutant SOD1 mice with a homozygous deletion of Plekhg5 takes at least 2 generations of breeding which can occupy more time than 9 months. To avoid this long period of crossbreeding, we analyzed the effect of Plekhg5 knockdown with two independent shRNA lentiviral constructs on embryonic motoneurons isolated from SOD1 G93A mutant mice *in vitro*. These new data show that knockdown of Plekhg5 in SOD1 G93A motoneurons results in markedly enhanced cell death (new Fig. 9). We also investigated ER-stress as a mayor pathogenic mechanism in SOD1 G93A motoneurons. PERK phosphorylation, BIP expression and Chop activation were enhanced in a supra-additive manner in SOD G93A motoneurons with additional Plekhg5 depletion. Interestingly, this effect is already observed in embryonic motoneurons representing a developmental stage long before disease becomes apparent *in vivo*. We did not observe any significant changes in the levels of transgenic SOD1 when Plekhg5 is depleted. This rules out the possibility that Plekhg5 is involved in the degradation of the mutant SOD protein. In summary, these data indicate that knockdown of Plekhg5 elevates the load of ER-stress in SOD1 G93A expressing cells.

Plekhg5-depletion in SOD1 G93A motoneurons results in elevated ER-stress. (e) SOD1 G93A - and non-transgenic motoneurons were depleted of Plekhg5 and the several ER-stress markers were examined after seven days in culture. (f) Quantification of Western blots shown in (e). (each data point represents one individual experiment; Mean \pm SEM; unpaired t-test; two-tailed).

Minor points:

1. In page 6, line 137, 'To assess whether dysregulated autophagy caused the accumulation at nerve terminals we examined the autophagic flux in Plekhg5 deficient motoneurons (Fig. 3).' Accumulation of what exactly are the authors referring to?

This experiment refers to accumulation of synaptic vesicle proteins. This information is now added in the revised version of our manuscript on page 8, line 241.

Reviewer #2 (Remarks to the Author):

Here, Luningschror et al, identifies the involvement of PLEKHG5 in regulating autophagy in motoneurons via its role as a GEF for Rab26, a small GTPase previously implicated in recruiting synaptic vesicles for autophagic degradation. The authors show that loss of PLEKHG5 results in the loss of motoneurons. In addition, membranous organelles appear in abnormally swollen neuromuscular junctions and is accompanied by an accumulation of synaptic vesicle proteins. Accordingly, motoneurons lacking PLEKH5 show diminished ability to form autophagosomes which would account for the unusual buildup of synaptic vesicle proteins. Overall the manuscript provides new and important insights towards our understanding of the processes governing turnover of synaptic vesicle proteins and is of importance for both targeted and general readerships.

Specific comments:

1. Authors mentioned that ultrastructurally membranes either showed open ends or failed to seal (Fig 2). Were these observations confirmed using serial section analyses?

We have not performed EM tomography and do not have data how these structures present at the 3D level. Our main focus was the morphology of synaptic vesicles as shown in Fig. 3 f and g, and we hope that classical EM analysis of these structures is sufficient to demonstrate the altered phenotype of synaptic vesicles when considered in context with the light microscopic and biochemical alterations. The additional structural alterations reflecting swollen axon terminals in 15 month old mice appear as part the neurodegenerative process which occurs as a consequence of disturbed autophagy. Similar structural observations have also been made in ALS-patient samples^{7,8}. Therefore, we have dampened our interpretation and replaced the corresponding description in the results sections to “At the ultrastructural level, axon terminals of *Plekhg5* deficient mice were filled with membrane fragments and cytoplasmic inclusions (Fig. 3), similar as those observed in patients^{7,8}. Strikingly, synaptic vesicles in *Plekhg5*^{-/-} motoneuron terminals were frequently enlarged, in contrast to synaptic vesicles in wildtype mice, which appeared smaller and of a more uniform size (Fig. 3 f - h on).” The revised paragraphs can be found on page 8, lines 162-166).

2. Acute overexpression of wildtype and constitutively active EGFP-Rab26 was previously shown to induce massive accumulation of vesicles containing synaptic vesicle proteins. Did the authors observe such accumulations in wildtype motoneurons? Do these accumulations diminish or perhaps even disappear in the PLEKHG5^{-/-} background with the overexpression?

We have performed additional experiments showing that wild type EGFP-Rab26 is found in vesicles within axons as previously described by Binotti et al⁹. In *Plekhg5* deficient cells, the size of these vesicles significantly decreases in axons, and also the number of heavily stained vesicles is markedly reduced (Fig. 8) in axons. When we overexpress constitutive active EGFP-Rab26, the number and size of these vesicles is normalized (Fig. 8), and this correlates with the normalization of axon growth observed in these motoneurons (Fig. 8).

Figure 8:

Expression of constitutive active Rab26 rescues axonal growth and autophagy defects in Plekhg5 deficient cells. (a) Representative images of GFP-Rab26-WT or GFP-Rab26-QL positive structures in axons of control cells and deficient Plekhg5 cells. (c, d) Size and number of axonal EGFP-Rab26 structures.

3. Experiments in Fig 4 elegantly detailed the role of PLEKHG5 as a Rab26 GEF. However, the subsequent experiments do not demonstrate the involvement of autophagy in this regard since measurements of axon growth were used as readout rather than parameters to monitor autophagy as were used in Fig 3. Furthermore, it is not immediately clear how restoration of a pathway involved in synaptic vesicle turnover results in recovery of neurite extension. Do synaptic terminals degenerate in these cultures as were observed in the animals that then possibly lead to neurite retraction/degeneration? Or could Rab26 itself be involved in trafficking vesicles for this extension?

In order to find out how altered Plekhg5/Rab26 activity leads to the axonal disease phenotype in Plekhg5 deficient mice we have performed additional experiments to characterize the morphology of axon terminals in Plekhg5^{-/-} motoneurons. Axon terminals of cultured Plekhg5^{-/-} motoneurons appear atrophic, their size is significantly smaller. Rab26 positive structures accumulate in cultured Plekhg5^{-/-} motoneurons in regions directly at active zones, in contrast to WT motoneurons where these structures are found in more proximal areas of axonal growth cones. This results in dysmorphic accumulation and clustering of synaptophysin positive synaptic vesicles (Fig 6 c) within the region where presynaptic active zones are located¹⁰, and also in altered actin structure. F-actin filaments appear reduced, as revealed by confocal analysis (Fig. 6 c). In order to investigate these morphological alterations with higher resolution, we performed Structured illumination microscopy. The corresponding results confirmed the data obtained by confocal analyses, and provided additional evidence that the number of Rab26 positive vesicles increased, and that these vesicles are coated with actin filaments. These

findings support the *in vivo* structural analyses shown in Fig. 4 b, c and Fig. 3 f, g demonstrating marked alterations in filamentous cytoskeletal structures. These data indicate that the accumulation of synaptic vesicles disturbs the structure of actin filaments, and this might cause the defect in axon growth.

Depletion of Plekhg5 results in axon growth defects and degeneration of axon terminals *in vitro*.

(c) Motoneurons were cultured for seven days and stained for synaptophysin, Rab26 and F-actin and imaged by confocal microscopy. (d) Quantification of axon terminal size. n = 15 cells. unpaired t-test. two-tailed. Axons of *Plekhg5^{-/-}* motoneurons display synaptophysin positive swellings.

In order to study effect of Plekhg5 deficiency on the fusion of autophagosomes with lysosomes in in proximal axons, we generated new lentiviral constructs which allowed us to follow the fate of retrogradely transported autophagosomes. We constructed lentiviral vectors that simultaneously express RFP-GFP-LC3 and Flag-Rab26-WT or Flag-Rab26-QL. We then quantified the number of RFP-GFP-LC3 positive structures in proximal axons and found that the number of retrogradely transported autophagosomes is reduced in *Plekhg5* deficient motoneurons. This reduction does not occur in *Plekhg5^{-/-}* motoneurons expressing constitutive active Rab26-QL. These data confirm our previous conclusion that Plekhg5 acts as a GEF for Rab26 and allows the generation of autophagosomes, which can fuse with lysosomes in proximal axons. We have included these new data in Figure 8.

Figure 8:

Expression of constitutive active Rab26 rescues axonal growth and autophagy defects in *Plekhg5* deficient cells. (f) Scheme of lentiviral vectors for simultaneous expression of RFP-GFP-LC3 and Flag-Rab26-WT or Flag-Rab26-QL. (g) Western blot analysis of Flag-Rab26-WT and Flag-Rab26-QL expression. (h) Motoneurons of *Plekhg5*^{+/+} and *Plekhg5*^{-/-} mice expressing mRFP-GFP-LC3 and Flag-Rab26-WT or Flag-Rab26-QL were cultured for seven days and the number mRFP-GFP-LC3 positive structures was analyzed. (i) Number of autophagosomes and autolysosomes upon expression of Flag-Rab26-WT and Flag-Rab26-QL. (three independent experiments with 15 cell analyzed in each experiment; Mean ± SEM; Two-way ANOVA; Bonferroni post-test).

4. Minor comment:

In Fig 2h, do the multiple lanes represent lysates from sciatic nerves obtained from different animals? If so, this should be mentioned in the figure legends.

It is as the reviewer suspects. This information is now included in the revised version of the figure legend on page 33, lines 959-960.

Reviewer #3 (Remarks to the Author):

The manuscript by Luningschror et al. describes the development and characterization of a *Plekhg5* knockout mouse. The authors demonstrate that the homozygous knockout mouse displays slowly progressive motor neuron degeneration, characterized by decreased survival, weakening grip strength, loss of motor neurons, alterations in muscle histology, and alterations in NMJ morphology. EM analysis indicates alterations in organelle morphology at synapses, and western blots demonstrate the accumulation of synaptic proteins. As these defects suggest a possible deficit in autophagy, the authors express a tandem GFP-RFP-LC3 probe and find deficits consistent with an impairment of autophagy. The authors hypothesize that *Plekhg5* may be a GEF for Rab26, and demonstrate that a fragment of *Plekhg5* acts as a GEF for Rab26 but not Rab5. They then go on to show that a constitutively active form of Rab26 rescues the axon growth defect seen in *Plekhg5*^{-/-} neurons.

The strength of this work is the generation and careful characterization of a mouse model for *Plekhg5* defects, and the demonstration of a defect in autophagy of synaptic components. This work makes a novel contribution, and should be published pending some important revisions:

Major:

1. The use of the constitutively active RAB26 construct to rescue the axon outgrowth deficit is a nice start, but the overall story would be much stronger if the authors showed that this construct also rescued the autophagy deficit observed in the *Plekhg5*^{-/-} neurons.

This comment overlaps with point 3 of reviewer 2. We have performed a series of additional experiments, as outlined in the response point 3 of reviewer 2, to go deeper in our analysis how *Plekhg5* and Rab26 signaling alter axon morphology and axon extension.

2. The colocalization of Rab26 and synaptophysin in Figs. 4 and 5S is not striking. These figures should be either strengthened, or de-emphasized – maybe the interaction is indirect or more likely transient.

Depletion of *Plekhg5* results in axon growth defects and degeneration of axon terminals *in vitro*.

(e) Motoneurons were cultured for seven days and stained for synaptophysin, Rab26 and F-actin and imaged by SIM microscopy.

We agree with the reviewer that it is difficult to draw conclusions about the identity of individual vesicles in axon terminals from doublestaining with Rab26 and synaptophysin on the light microscopic level. The major question we wanted to address with these experiments is whether accumulated vesicles observed by EM analyses express Synaptophysin or Rab26. In order to increase the resolution of the immunofluorescence analyses on the light microscopic level, we performed SIM microscopy and confirmed that Synaptophysin positive vesicles accumulate in presynaptic structures and that some of these vesicles also express high levels of Rab26. We also followed the advice of the reviewer to de-emphasize these conclusions from these data. The new data are now included as Fig. 6 e and discussed in the revised text on pages 11, lines 256-261.

3. Is it sufficient to show specificity by comparing the GEF activity of Plekhg5 between only two Rabs? Most studies perform multiple comparisons. At the least, Rab7 and Rab11 should also be assayed.

We have now complemented this data set with Rab 27b and Rab33b. In both cases Plekhg5 did not show any GEF activity, supporting the idea that Plekhg5 is specific for Rab26. The new data are now included in the new Fig. 7 e, f.

Plekhg5 functions as a GEF for Rab26. (d - h) Exchange activity of Plekhg5 on different GTPases. Exchange activity of Plekhg5 on Rab5 (d), Rab27b (e), Rab33b (f) and Rab26 (g) was measured by monitoring the fluorescent increase of Mant-GppNhp upon binding to Rab proteins. (g) The initial GTP exchange rate of Rab26 was evaluated using the first 10 s of each time course. (Mean \pm SEM; Student's t-test; two-tailed)

4. The blots shown in Fig. 3d are perplexing, because most labs see a robust LC3-I/LC3-II doublet, rather than a single major band for LC3-II. Can the authors include a parallel positive control, brain extract for example, showing the usual doublet? And can they comment on this unusual observation?

We conducted LC3 Western blots using spinal cord lysates and detected a robust LC3-I/LC3-II doublet as usually observed and mentioned by the reviewer. The prominent LC3-I band observed in Fig. 5 g,h probably reflects the fast autophagic flux upon treatment with Bafilomycin A in the neurosphere-derived neurons that have been used for this experiment.

Plekhg5 regulates biogenesis of autophagosomes. (i) Western blot analysis of LC3 expression in spinal cord extracts. (h) Quantification of LC3 Western blots from spinal cord extracts. (Mean ± SEM; n = 4; Mean ± SEM; student's t-test; two-tailed).

5. The description of the interaction of Plekhg5 with PI(3) seems minimal. Further, the logical progression leading to the conclusion described in lines 179-180 seems to be missing some key steps.

We have now expanded the description of this experiment on pages 9-10, lines 222-241. The low interaction of Plekhg5 with PI(3)P fits to the hypothesis that Plekhg5 is a cytosolic protein which is recruited to preautophagosomal membranes to act as a GEF and then becomes degraded when these vesicles with activated Rab26 are fused with lysosomes. Therefore, only a minor fraction of Plekhg5 interacts with PI(3)P as observed in the pulldown assay.

6. The presentation of the work is too succinct, lacks important information, and overstates a number of points. For example, the abstract includes broad statements that are not yet fully supported, rather than discussing the specific observations made in the paper, and their interesting and important implications. The introduction does not provide a useful summary of what is known about the links between Plekhg5 and disease. The body of the manuscript lacks a schematic of Plekhg5 that clarifies which domains were used in the biochemical experiments presented. The summary section very broadly discusses links between autophagy defects and disease, more suitable to a broad-based review, rather than discussing the observations made in this work and their more specific implications. And as noted below, a number of figures in the supplement would be more helpful if they appeared within the main text.

We have revised the text and extended the introduction (pages 4-5, lines 75-96) and the discussion (pages 14-15, lines 337-374) to match with our new data and included a scheme in Fig. 7 c, which clarifies which domains of Plekhg5 have been used for the biochemical experiments. We have also moved the requested Suppl. Figure into the main part and extended the number of Figures from 4 to 9.

Minor:

1. Both the grip strength data from Fig. S2a and the staining data from Fig. S5 should be incorporated in the main text. If necessary to save space, Fig. 1d could be relocated to the supplement.

As the reviewer suggested, we incorporated both data sets in the main figures. The extended grip strength data can now be found in Fig. 1 e, f. The staining data are now present in Fig. 6 c.

2. Is there any way to provide a quantitative analysis of the muscle histology shown in Fig. 1e?

We have now complemented the phenotypic characterization of the motor system with electromyography recordings of the gastrocnemius and plantaris muscles providing quantitative data of muscle de- and reinnervation. These new data are shown in figure 1 i-l.

3. Initial fusion between autophagosomes and LAMP1-positive late endosomes is generally thought to occur distally in the axon, while full maturation to an autolysosome may only occur more proximally (lines 151-155).

We agree with the reviewer. This part of the text referred to acidification of autolysosomes, as a last step of maturation, which occurs in proximal axons. We revised the text accordingly on page 9, lines 202-204.

4. Nutrient deprivation differentially affects neurons and cell lines. Therefore, the Plekhg5 degradation experiments performed in NSC34 cells should be repeated in primary neurons.

As suggested by the reviewer, we have repeated this experiment with primary motoneurons and received similar results as with NSC34 cells that starvation by nutrient deprivation leads to degradation of Plekhg5. The identity of the bands detected by Western blot was controlled by shRNA- mediated knockdown of Plekhg5.

Plekhg5 regulates biogenesis of autophagosomes. (I) Cultured motoneurons were cultured for seven days in motoneuron-media. At day seven cells were cultured for 4 h with different media each deprived from specific components. Subsequently, motoneurons were lysed and the expression of endogenous Plekhg5 was analyzed by Western blot. Asterisks label bands that do not change upon nutrient deprivation. Arrowheads point to bands that change upon nutrient deprivation.

In summary, this is interesting, novel work that will be of high interest to those in the field. However, a number of key revisions are required prior to publication.

References:

1. Schmalbruch, H., Jensen, H.J., Bjaerg, M., Kamieniecka, Z. & Kurland, L. A new mouse mutant with progressive motor neuronopathy. *J Neuropathol Exp Neurol* **50**, 192-204 (1991).
2. Gurney, M.E. *et al.* Motor neuron degeneration in mice that express a human Cu,Zn superoxide dismutase mutation. *Science* **264**, 1772-1775 (1994).
3. Tankersley, C.G., Haenggeli, C. & Rothstein, J.D. Respiratory impairment in a mouse model of amyotrophic lateral sclerosis. *J Appl Physiol (1985)* **102**, 926-932 (2007).
4. Newrick, P.G. & Langton-Hewer, R. Pain in motor neuron disease. *Journal of neurology, neurosurgery, and psychiatry* **48**, 838-840 (1985).
5. Chio, A., Mora, G. & Lauria, G. Pain in amyotrophic lateral sclerosis. *Lancet Neurol* **16**, 144-157 (2017).
6. Ince, P.G. Neuropathology, in *Amyotrophic Lateral Sclerosis*, Vol. Handbook, Edn. First Edition. (ed. E.R.H.M. Brown Jr, V.; Swash, M.) (Dunitz, Martin, United Kingdom; 2000).
7. Chou, S.M. Pathology-Light Microscopy of Amyotrophic Lareral Sclerosis, in *Handbook of Amyotrophic Lareral Sclerosis*. (ed. R.A. Smith) (1992).
8. Bruneteau, G. *et al.* Endplate denervation correlates with Nogo-A muscle expression in amyotrophic lateral sclerosis patients. *Ann Clin Transl Neurol* **2**, 362-372 (2015).
9. Binotti, B. *et al.* The GTPase Rab26 links synaptic vesicles to the autophagy pathway. *Elife* **4**, e05597 (2015).
10. Jablonka, S., Beck, M., Lechner, B.D., Mayer, C. & Sendtner, M. Defective Ca²⁺ channel clustering in axon terminals disturbs excitability in motoneurons in spinal muscular atrophy. *The Journal of cell biology* **179**, 139-149 (2007).

REVIEWERS' COMMENTS:

Reviewer #1 (Remarks to the Author):

In their revised manuscript, Luningschor addressed all my questions and suggestions providing a detailed point-by-point response. The study is very interesting to the field of neurodegeneration, particularly to motoneuron disease, and suitable now for publication in Nature Communications. I recommend acceptance of the manuscript without further revision.

Reviewer #2 (Remarks to the Author):

Here, Luningschor and coauthors present a revised and significantly improved version of their previous manuscript reporting on the involvement of a Plekhhg5-mediated pathway that is potentially implicated in the pathophysiology of motor neuron disorders. The authors have addressed most of the issues raised in the previous submission and I have no further major concerns about the manuscript.

A minor point to note is that although the authors mentioned the use of FLAG-tagged versions of Rab26 in Figure 8 (f-i), the label "EGFP-Rab26" was used in the corresponding panels for (h) and (i) - I wonder if this could be typographical errors? Also, it would be nice to include the corresponding staining for FLAG-Rab26 in these neurons to show the spatial distribution of RAB26 in relation to the autophagosomes.

Reviewer #3 (Remarks to the Author):

The authors have done an excellent job replying to the points raised by all three referees, and I continue to be enthusiastic about the work as a whole.

I am a little confused about one point that was not addressed as thoroughly in the revision - the relatively limited number of Rabs tested. These data do not allow the authors to conclude specificity, so they should carefully review their statements on this point to ensure that they are not overstating their observations.

In all other respects, I am fully satisfied and look forward to seeing this work published.

REVIEWERS' COMMENTS:

Reviewer #1 (Remarks to the Author):

In their revised manuscript, Luningschor addressed all my questions and suggestions providing a detailed point-by-point response. The study is very interesting to the field of neurodegeneration, particularly to motoneuron disease, and suitable now for publication in Nature Communications. I recommend acceptance of the manuscript without further revision.

We thank reviewer #1 for her/his comments and suggestions.

Reviewer #2 (Remarks to the Author):

Here, Luningschor and coauthors present a revised and significantly improved version of their previous manuscript reporting on the involvement of a Plekhg5-mediated pathway that is potentially implicated in the pathophysiology of motor neuron disorders. The authors have addressed most of the issues raised in the previous submission and I have no further major concerns about the manuscript.

A minor point to note is that although the authors mentioned the use of FLAG-tagged versions of Rab26 in Figure 8 (f-i), the label "EGFP-Rab26" was used in the corresponding panels for (h) and (i) - I wonder if this could be typographical errors? Also, it would be nice to include the corresponding staining for FLAG-Rab26 in these neurons to show the spatial distribution of RAB26 in relation to the autophagosomes.

We thank the reviewer for making us aware that Figure 8 (f-i) was mislabeled. We changed the labeling to "Flag-Rab26".

The staining of neurons with FLAG-Rab26 has been a central part of the study by Binotti et al., eLIFE 2016. In the previous paper, the spatial distribution of Rab26 in relation to different autophagosomal markers has already been investigated in detail. Therefore, in order not to overload our study with data that in principle already exist, we have not included such data.

Reviewer #3 (Remarks to the Author):

The authors have done an excellent job replying to the points raised by all three referees, and I continue to be enthusiastic about the work as a whole.

I am a little confused about one point that was not addressed as thoroughly in the revision - the relatively limited number of Rabs tested. These data do not allow the authors to conclude specificity, so they should carefully review their statements on this point to ensure that they are not overstating their observations.

In all other respects, I am fully satisfied and look forward to seeing this work published.

We also thank reviewer #3 for her/his comments and suggestions. Our data show that Plekhg5 acts as a GEF for Rab26, but not for Rab5, Rab33b and Rab27b. We agree with the reviewer that these data do not allow us to claim that Plekhg5 is specifically functioning as a GEF for Rab26. It was not the intention of our manuscript to stress this point. On the other hand we can conclude that Plekhg5 is not acting on all Rab proteins in general, pointing to relative specificity within this group of small GTPases. We have changed our manuscript according the suggestion made by the reviewer.